# PfCRT mutations conferring piperaquine resistance in falciparum malaria shape the kinetics of quinoline drug binding and transport

**Guillermo M. Gomez**[1], **Giulia D'Arrigo**[2], **Cecilia P. Sanchez**[1], **Fiona Berger**[1], **Rebecca C. Wade**[2,3,4], **Michael Lanzer**[1] *

**1** Center of Infectious Diseases, Parasitology, Universitätsklinikum Heidelberg, Im Neuenheimer Feld, Heidelberg, Germany, **2** Molecular and Cellular Modelling Group, Heidelberg Institute for Theoretical Studies (HITS), Schloss-Wolfsbrunnenweg, Heidelberg, Germany, **3** Center for Molecular Biology (ZMBH), DKFZ-ZMBH Alliance, Heidelberg University, Im Neuenheimer Feld, Heidelberg, Germany, **4** Interdisciplinary Center for Scientific Computing (IWR), Heidelberg University, Im Neuenheimer Feld, Heidelberg, Germany

* michael.lanzer@med.uni-heidelberg.de

**Data Availability Statement:** The data that support the findings of this study are publicly available from Dryad repository and the Zenodo repository with

## Abstract

The chloroquine resistance transporter (PfCRT) confers resistance to a wide range of quinoline and quinoline-like antimalarial drugs in *Plasmodium falciparum*, with local drug histories driving its evolution and, hence, the drug transport specificities. For example, the change in prescription practice from chloroquine (CQ) to piperaquine (PPQ) in Southeast Asia has resulted in PfCRT variants that carry an additional mutation, leading to PPQ resistance and, concomitantly, to CQ re-sensitization. How this additional amino acid substitution guides such opposing changes in drug susceptibility is largely unclear. Here, we show by detailed kinetic analyses that both the CQ- and the PPQ-resistance conferring PfCRT variants can bind and transport both drugs. Surprisingly, the kinetic profiles revealed subtle yet significant differences, defining a threshold for *in vivo* CQ and PPQ resistance. Competition kinetics, together with docking and molecular dynamics simulations, show that the PfCRT variant from the Southeast Asian *P. falciparum* strain Dd2 can accept simultaneously both CQ and PPQ at distinct but allosterically interacting sites. Furthermore, combining existing mutations associated with PPQ resistance created a PfCRT isoform with unprecedented non-Michaelis-Menten kinetics and superior transport efficiency for both CQ and PPQ. Our study provides additional insights into the organization of the substrate binding cavity of PfCRT and, in addition, reveals perspectives for PfCRT variants with equal transport efficiencies for both PPQ and CQ.

## Author summary

Chloroquine (CQ) used to be the drug of choice against malaria until the parasite responsible for the disease became resistant. In the 1970s, piperaquine (PPQ) was introduced in areas where resistance to CQ was wide spread. In the following decade, an estimated 140

the identifier(s) https://doi.org/10.5061/dryad.
prr4xgxr5 and https://doi.org/10.5281/zenodo.
7900741, respectively. In addition, the relevant
experimental data are within the manuscript and its
Supporting Information files.

**Funding:** This work was funded by the institutional
support from the State of Baden-Württemberg to
ML. RCW thank the Klaus Tschira Foundation and
the European Union's Horizon 2020 Framework
Programme for Research and Innovation under the
Specific Grant Agreement No. 945539 (Human
Brain Project SGA3) for support. The funders had
no role in study design, data collection and
analysis, decision to publish, or preparation of the
manuscript.

**Competing interests:** The authors have declared
that no competing interests exist.

million doses were distributed, which substantially reduced the malaria burden, particularly in China, but created an environment in which PPQ resistant strains of the human malaria parasite *Plasmodium falciparum* emerged and spread. Interestingly, the PPQ resistant parasites displayed an increased CQ sensitivity. The main genetic determinant of both CQ and PPQ resistance in *P. falciparum* is a drug transporter, termed PfCRT. In this study, we used biochemical and bioinformatics approaches to understand how mutational changes in PfCRT influence the interaction of the carrier with CQ and PPQ. We found that PfCRT from CQ resistant parasites is better at transporting CQ than are PfCRT variants from PPQ resistant parasites, while the opposite is true for PPQ. We also found that PfCRT can be engineered such that it transports both antimalarials equally well. Our study offers insights into how PfCRT has evolved in response to changing drug pressure, and raises concerns regarding how it may evolve in the future.

## Introduction

Emerging drug resistance and a disruption of medical services during the corona pandemic threaten recent gains in malaria control and have led to an increase in disease episodes and mortality by 6% and 9%, respectively, since 2019 to an estimated 247 million cases and ~619,000 deaths as of 2021 [1]. A hotspot for drug resistance is the Greater Mekong subregion, where indiscriminate drug use and migration of seasonal workers between low and high transmission areas have created a breeding ground for drug tolerant malaria [2]. The first drug to fail was chloroquine (CQ) in the late 1950s, followed by piperaquine (PPQ) in the 1980s, artemisinin in 2008 and dihydroartemisinin/piperaquine combination therapy in 2012 [2–4].

Drug resistance particularly affects the most dangerous and most frequent form of human malaria caused by the protozoan parasite *Plasmodium falciparum*. Continuous but changing drug exposure selected for *P. falciparum* strains harboring multiple drug resistant traits, together with an accelerated mutator phenotype, which is thought to allow for quick adaption to changing drug environments [5,6]. This has raised concerns of an incurable multi-drug resistant "superbug". However, the ground truth is more complex, as exemplified by quinoline antimalarials that, partnered with artemisinin derivatives, comprise the mainstay of malaria chemotherapy. While there is cross-resistance within the quinoline drug class, e.g., between CQ, quinine and amodiaquine, other members show a re-sensitizing behavior [5,7]. This re-sensitizing phenotype is best documented for CQ and PPQ, with parasites becoming more CQ sensitive upon PPQ selection [8–10]. A better understanding of the molecular mechanisms favoring cross-resistance or re-sensitization might help extend the longevity and efficacy of currently deployed drugs and inform on criteria for next generation quinoline antimalarials.

The main driver of resistance to quinoline and quinoline-like anti-malaria drugs is the CQ resistance transporter, PfCRT [5,11,12]. PfCRT features 10 transmembrane domains, two re-entrant loops and an internal pseudo-symmetry, and belongs to the drug/metabolite transporter family [9,13]. It resides at the membrane of the parasite's digestive vacuole, an acidic proteolytic organelle, in which hemoglobin endocytosed by the parasite during intra-erythrocytic development is degraded to oligopeptides [14]. The proteolysis of approximately 80% of the red blood cell hemoglobin provides nutrients and space for the anabolic activities of the parasite and, in addition, contributes to maintaining the colloid/osmotic balance of the infected erythrocyte [15,16]. To feed into the metabolic circuits of the parasite, the oligopeptides must cross the digestive vacuolar membrane into the parasite's cytoplasm. Recent

developments suggest that PfCRT exercises this function, whereby PfCRT seems to accept a broad range of neutral and single charged oligopeptides of 4–11 residues in length [17,18].

Quinoline antimalarial drugs target the digestive vacuole by interfering with the biomineralization of toxic heme liberated as a byproduct of hemoglobin degradation to inert hemozoin [19,20]. To neutralize the action of quinoline drugs, the parasite expels them from the digestive vacuole in a process that is mediated by mutated PfCRT [21–24]. Polymorphic PfCRT variants can carry 4 to 10 mutational changes, including a conserved K76T replacement, with geographic origin and drug history defining the type and number of acquired amino acid replacements [5,7,25]. For example, the dominant PfCRT variant in Southeast Asia, termed PfCRT$^{Dd2}$, harbors 8 mutational changes [12]. PfCRT$^{Dd2}$ mediates resistance to CQ and it modulates the responsiveness to quinine and amodiaquine, among other compounds [5,25]. An additional 9$^{th}$ amino acid replacement, such as H97Y, F145I, M343L or G353V, confers PPQ resistance at the expense of an increasing responsiveness to CQ [8,9,26]. How these substitutions shape the structure and function of PfCRT is only partly understood.

The structure of PfCRT has recently been resolved at 3.2 Å resolution in the open-to-vacuole conformation, revealing a large substrate binding cavity formed by an antiparallel arrangement of transmembrane domains TM1-TM4 and TM6-TM9 [9]. However, a structural-functional description of the substrate binding cavity is still lacking. In the absence of structural information regarding PfCRT-substrate complexes, competition kinetics can distinguish between different mechanisms of substrate binding and, hence, can provide insights into the organization of an uncharted binding cavity. In this context, we have recently demonstrated that PfCRT can accept CQ and quinine at separate but interdependent sites in a partial mixed-type inhibition process [22].

Here, we have probed the substrate binding cavity of PfCRT using CQ and PPQ as a model for a re-sensitizing drug combination. Our data show that both PfCRT$^{Dd2}$ and any of the PfCRT variants mediating PPQ-resistance are fully capable of transporting both drugs, albeit with distinct kinetic properties. Furthermore, CQ and PPQ occupy separate regions within the substrate binding cavity, as shown by competition kinetics and molecular dynamics (MD) simulations. Finally, we demonstrate that re-shuffling of amino acid replacements can create PfCRT variants with improved transport efficiencies at higher drug concentrations and non-Michaels-Menten kinetics for both CQ and PPQ, suggesting that the currently observed re-sensitizing phenomenon represents only a snapshot in the evolution of PfCRT.

## Results

### Subtle differences in transport kinetics characterize PPQ resistance conferring PfCRT variants

Fig 1 depicts the time courses of radio-labeled CQ or PPQ uptake from an external concentration of 50 μM by *Xenopus laevis* oocytes expressing the PfCRT variant from the CQ resistant *P. falciparum* strain Dd2 (PfCRT$^{Dd2}$) or the related PfCRT$^{Dd2\_F145I}$, the latter carrying an additional mutation (F145I) associated with PPQ resistance and CQ re-sensitization. The F145I mutation was initially chosen because it confers a high degree of PPQ resistance [8,27,28]. Water-injected oocytes and oocytes expressing the wild type PfCRT isoform from the CQ sensitive *P. falciparum* strain HB3 were investigated in parallel to account for intrinsic drug accumulation. Oocytes expressing PfCRT$^{Dd2}$ or PfCRT$^{Dd2\_F145I}$ took up significantly more CQ and PPQ than did water-injected and PfCRT$^{HB3}$-expressing control oocytes (Fig 1A and 1B, left panels) (PfCRT$^{Dd2}$: $p$ <0.001, df = 8, F = 74.59; $p$ <0.05, df = 8, F = 4.61; for CQ and PPQ, respectively; PfCRT$^{Dd2\_F145I}$: $p$ <0.001, df = 8, F = 26.58; $p$ <0.005, df = 8, F = 15.17; for CQ and PPQ, respectively; F-test). Apparently, both PfCRT$^{Dd2}$ and PfCRT$^{Dd2\_F145I}$ are capable of transporting CQ and PPQ, yet

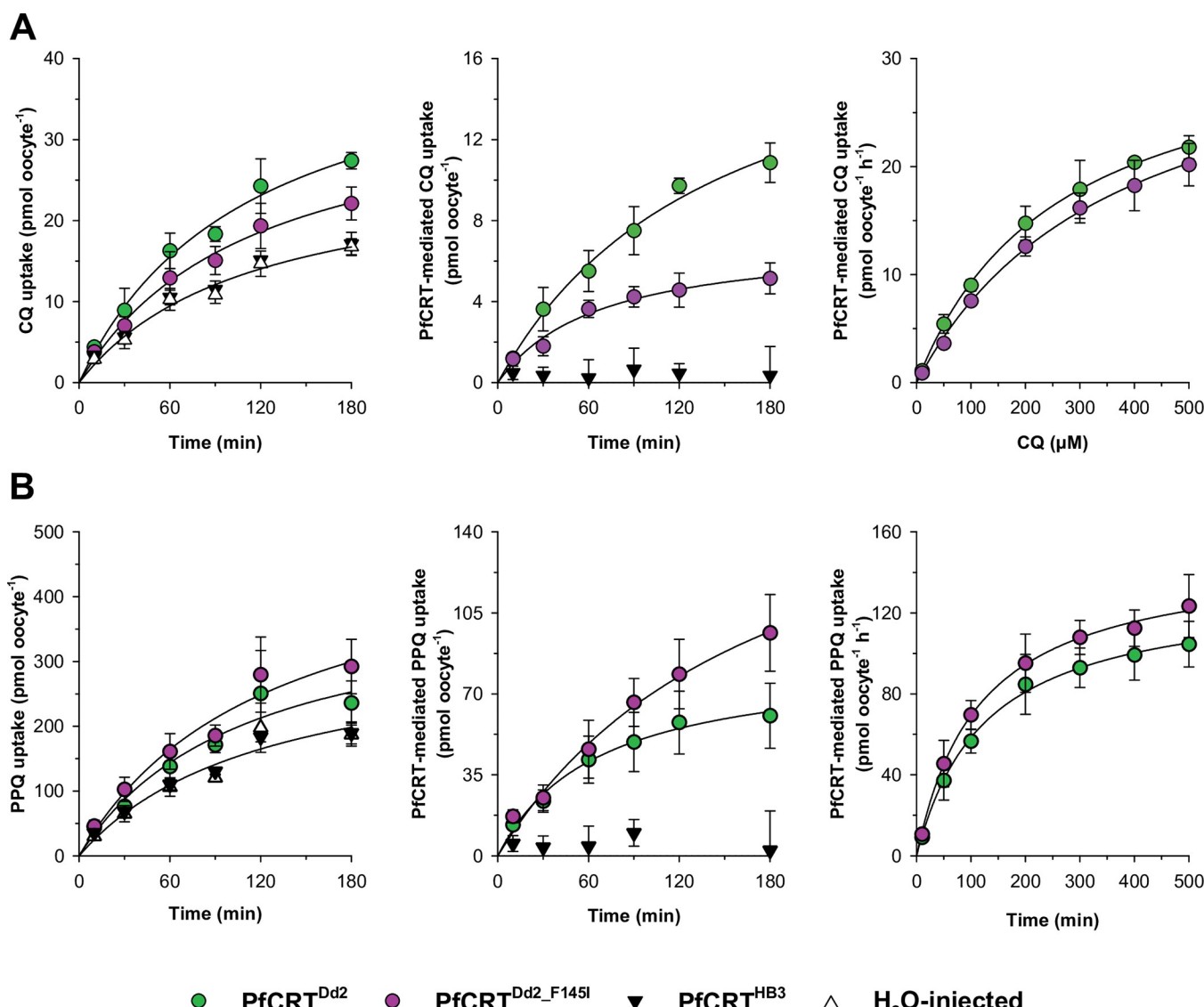

**Fig 1. Effect of the F145I substitution on PfCRT$^{Dd2}$-mediated drug transport.** A, B. PfCRT$^{Dd2}$ and PfCRT$^{Dd2\_F145I}$ mediate CQ and PPQ uptake, respectively. *Left panels*, time course for the uptake of CQ (A) and PPQ (B) into water-injected oocytes (open triangles) and oocytes expressing PfCRT$^{Dd2}$ (green circles), PfCRT$^{HB3}$ (filled triangles) or PfCRT$^{Dd2\_F145I}$ (purple circles) from extracellular drug concentrations of 50 μM (radiolabeled plus unlabeled, extracellular pH 6.0). *Middle panels*, PfCRT-mediated drug uptake as calculated by subtracting uptake in water-injected oocytes from that measured in PfCRT-expressing oocytes at each time point. *Right panels*, concentration dependence of the PfCRT-mediated uptake of CQ (A) and PPQ (B) over extracellular drug concentrations ranging from 10 to 500 μM (radiolabeled plus unlabeled). A least-squares fit of the Michaelis-Menten equation to the resulting data yielded the kinetic parameters summarized in Table 1. In all cases, the mean ± SEM of at least 3 independent biological replicates are shown, whereby a biological replicate corresponds to measurements of at least 10 oocytes per condition.

with opposite drug preferences. Whereas PfCRT$^{Dd2}$ transported more CQ per time unit than did PfCRT$^{Dd2\_F145I}$, the opposite was true for PPQ, with PfCRT$^{Dd2\_F145I}$ transporting more PPQ per time unit than did PfCRT$^{Dd2}$ (Fig 1A and 1B, middle panels).

A detailed kinetic analysis, based on the uptake of CQ or PPQ from increasing external concentrations (10, 50, 100, 200, 300, 400 or 500 μM) at the 60 min time point (Fig 1A and 1B, right panels), revealed saturation kinetics and explained the differential transport rates displayed by the two PfCRT forms by different values for the Michaelis-Menten constant $K_M$ and the maximal transport velocity $V_{max}$. The two kinetic parameters were derived by fitting the Michaelis-Menten

**Table 1. Kinetic parameters of CQ and PPQ transport mediated by various PfCRT isoforms.** Kinetic parameters were obtained from Figs 1, 2 and 10. $V_{max}$, maximal velocity of drug transport; $K_M$, Michaelis constant; $H_c$, Hill coefficient; N, number of determinations per kinetic analysis; n, total number of oocytes.

| Substrate | Parameter | PfCRT$^{Dd2}$ | PfCRT$^{Dd2\_F145I}$ | PfCRT$^{Dd2\_H97Y}$ | PfCRT$^{Dd2\_M343L}$ | PfCRT$^{Dd2\_G353V}$ | PfCRT$^{Dd2\_H97Y\_F145I}$ |
|---|---|---|---|---|---|---|---|
| CQ | $V_{max}$ (pmol oocyte$^{-1}$ h$^{-1}$) | 33 ± 1 | 35 ± 2 | 24 ± 1 | 31 ± 2 | 23 ± 3 | 35 ± 4 |
| | $K_M$ (μM) | 260 ± 10 | 370 ± 30 | 184 ± 16 | 530 ± 60 | 270 ± 80 | 240 ± 40 |
| | $H_c$ | | | | | | 1.6 ± 0.2 |
| | N, n | 12, 195 | 6, 108 | 7, 123 | 7, 86 | 8, 83 | 9, 139 |
| PPQ | $V_{max}$ (pmol oocyte$^{-1}$ h$^{-1}$) | 131 ± 3 | 149 ± 3 | 131 ± 6 | 163 ± 10 | 134 ± 10 | 131 ± 10 |
| | $K_M$ (μM) | 125 ± 9 | 115 ± 7 | 106 ± 14 | 80 ± 16 | 90 ± 20 | 170 ± 20 |
| | $H_c$ | | | | | | 2.0 ± 0.3 |
| | N, n | 9, 92 | 14, 140 | 10, 133 | 11, 95 | 13, 99 | 10, 154 |

equation to the kinetic data (Table 1). It was found that the F145I substitution increased the $K_M$ for CQ by 44% (from 260 ± 10 μM for PfCRT$^{Dd2}$ to 370 ± 30 μM; $p = 0.004$; df = 10; F = 14.43, F-test), without affecting the $V_{max}$ ($p = 0.259$; df = 10; F = 1.43, F-test). Conversely, the F145I substitution increased the $V_{max}$ for PPQ by 14% (from 131 ± 3 pmol oocyte$^{-1}$ h$^{-1}$ for PfCRT$^{Dd2}$ to 149 ± 3 pmol oocyte$^{-1}$ h$^{-1}$; $p = 0.003$; df = 10; F = 15.05, F-test), but did not affect the $K_M$ ($p = 0.387$; df = 10; F = 0.82, F-test). The kinetic parameters determined for CQ transport by PfCRT$^{Dd2}$ were comparable with previous measurements [21,22].

To test our finding that mutations associated with PPQ resistance improve the transport kinetics for PPQ while worsening that of CQ, we extended our study to include three additional PfCRT variants (Fig 2A and 2B). Interestingly, the mutations associated with PPQ

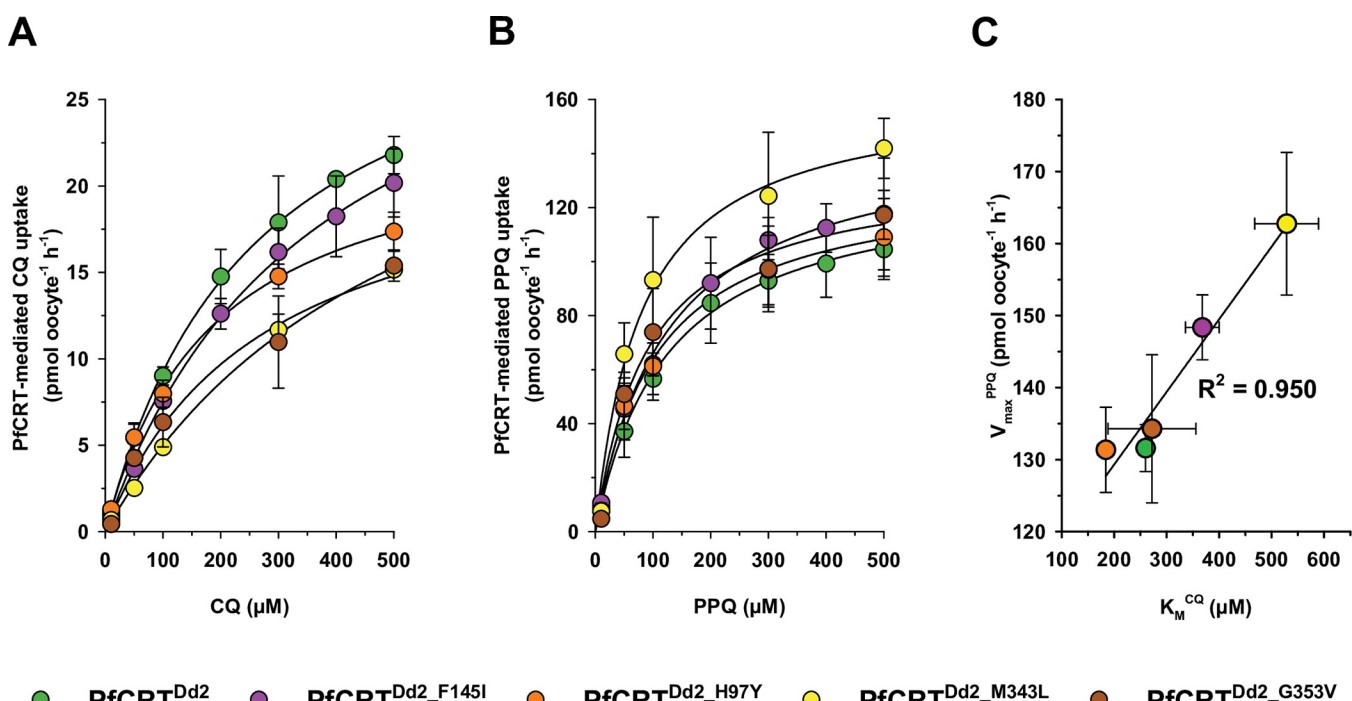

**Fig 2. Kinetics of CQ and PPQ transport by various PfCRT isoforms associated with PPQ resistance.** A, B. The specific uptake of CQ (A) and PPQ (B) was measured over an extracellular concentration range of 10 to 500 μM (radiolabeled plus unlabeled), at pH 6, into oocytes expressing PfCRT$^{Dd2}$ (green), PfCRT$^{Dd2\_F145I}$ (purple), PfCRT$^{Dd2\_H97Y}$ (orange), PfCRT$^{Dd2\_M343L}$ (yellow) or PfCRT$^{Dd2\_G353V}$ (brown). The respective kinetic parameters are presented in Table 1. The mean ± SEM of at least 3 independent biological replicates are shown per condition. C. Correlation analysis between the PPQ $V_{max}$ and the CQ $K_M$ data were taken from Table 1. The $R^2$ value is indicated.

resistance can affect either the $K_M$ or the $V_{max}$ or both values for both CQ and PPQ (Table 1). For example, the H97Y mutation resulted in a significantly lower $K_M$ value for CQ and a lower $K_M$ value for PPQ ($p = 0.005$ and $p = 0.273$, respectively). It also led to a reduced $V_{max}$ for CQ, but unchanged $V_{max}$ for PPQ, as compared with PfCRT$^{Dd2}$ ($p < 0.001$ and $p = 0.978$, respectively). In comparison, the M343L mutation combined a higher $K_M$ value for CQ with an unaltered CQ $V_{max}$ and a decreased PPQ $K_M$ with an increased $V_{max}$ for PPQ ($p < 0.001$, $p = 0.403$, $p = 0.071$, and $p = 0.020$, respectively). The G353V mutation significantly lowered the $V_{max}$ for CQ and moderately reduced the $K_M$ for PPQ, while maintaining both the $K_M$ for CQ and $V_{max}$ for PPQ of PfCRT$^{Dd2}$ ($p = 0.023$, $p = 0.185$, $p = 0.855$, and $p = 0.791$, respectively). Plotting the PPQ $V_{max}$ values as a function of the CQ $K_M$ values revealed a linear correlation between the two kinetic parameters (Fig 2C). Apparently, a higher PPQ transport rate comes at the cost of a lower affinity for CQ.

Immunofluorescence microscopy and semi-quantitative Western analyses, using a guinea pig antiserum against PfCRT and appropriate co-localization and loading controls, revealed comparable expression levels of the PfCRT variants in the oocyte system, with the transporters being localized at the oolemma (S1 and S2 Figs). Water-injected oocytes did not react with the PfCRT antisera. As a control for the specificity of the PfCRT-mediated transport rates, we evaluated the responsiveness to verapamil, a partial mixed-type inhibitor of PfCRT [22]. In all cases, the PfCRT-mediated CQ and PPQ transport was inhibited in the presence of 100 μM verapamil (S3 Fig).

## CQ and PPQ bind at distinct sites

A recent study has suggested that PfCRT, as exemplified by PfCRT$^{7G8}$, can accept either CQ or PPQ in a competitive binding model [9], which would imply that both drugs bind to the same or at least overlapping binding sites. To test this hypothesis, we performed competition plot experiments, in which the enzyme or carrier is exposed to substrate mixtures that give the same transport rate for different substrates [29]. The experiment is performed with substrate A labeled and then repeated with substrate B labeled. The combined transport velocities, $V_{total}$, is subsequently analyzed as a function of the proportion of substrate A in the mixture, P. If the two substrates bind to the same site, then $V_{total}$ is independent of P and the competition plot shows a horizontal line [29]. In the case of PfCRT$^{Dd2}$ and the two substrates CQ and PPQ, the competition plot gives a curve with a maximum, indicating that CQ and PPQ bind at different sites (Fig 3A). Similar results were found for PfCRT$^{Dd2\_F145I}$. Again, $V_{total}$ was dependent on P, with the curve revealing a concave relationship (Fig 3B). These findings refute a competitive binding model and, instead, point towards an allosteric binding model in which CQ and PPQ occupy different domains within the substrate binding cavity of PfCRT$^{Dd2}$ and PfCRT$^{Dd2\_F145I}$.

To further characterize the kinetic relationship between CQ and PPQ, we performed a series of competition reactions. To this end, water-injected oocytes and PfCRT-expressing oocytes were incubated in medium containing one of seven concentrations of radio-labeled CQ and one of six concentrations of cold PPQ for 60 min before the amount of radio-labeled CQ uptake was determined. The rate of PfCRT-mediated CQ uptake was calculated for each condition by subtracting the corresponding value obtained for water-injected oocytes from those of PfCRT-expressing oocytes. The results were analyzed as a function of the CQ concentration (Fig 4A) and 16 different models of substrate competition were globally fit to the data by the least-squares method, using Python. The models were ranked, according to their Akaike information criterion difference ($\Delta AIC_C$) and their Akaike weight (Fig 4B), with the most plausible model being the one with the lowest $AIC_C$ ($\Delta AIC_C = 0$) and the highest Akaike weight [30].

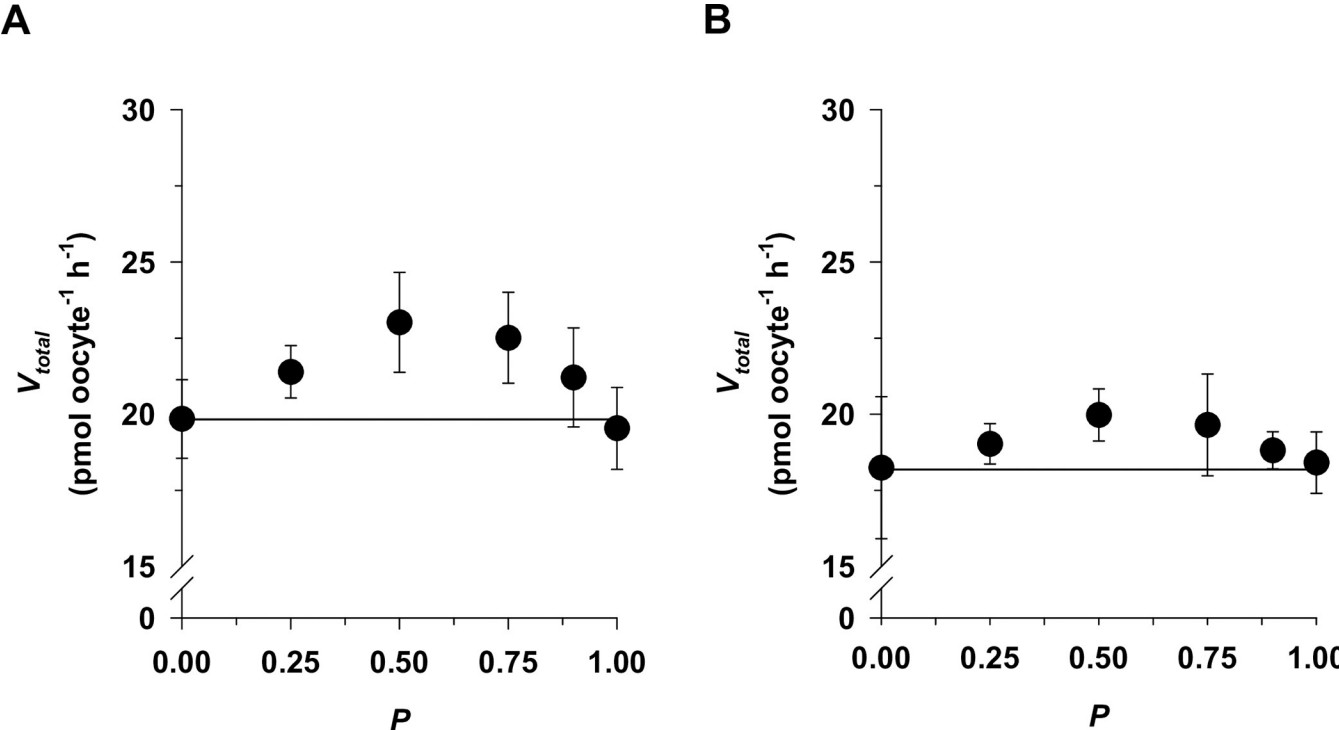

**Fig 3.** Competition plot of the interaction of PfCRT$^{Dd2}$ (A) and PfCRT$^{Dd2\_F145I}$ (B) with CQ and PPQ. The competition plot tests whether two substrates bind to the same site or distinct sites on an enzyme or transporter [29]. A competitive reaction would yield data points that lie on a horizontal line, whereas allosteric interactions would reveal a curved relationship. PfCRT-mediated drug uptake was measured from mixtures of radiolabeled CQ and PPQ that combined yielded an uptake close to saturation ($V_{total}$). $V_{total}$ was subsequently analyzed as a function of the proportion of PPQ in the mixture (P). For more details see Materials and Methods. The mean ± SEM of at least 3 independent biological replicates are shown.

The two models that best described the interaction between CQ, PPQ and PfCRT$^{Dd2}$ were partial noncompetitive inhibition and partial mixed-type inhibition (Fig 4B). To discriminate between these two models, we performed an F-test. The F-test statistically favored the partial noncompetitive inhibition model ($p = 0.886$, F = 0.021, df = 37). The kinetic parameters defining the partial noncompetitive and the partial mixed-type inhibition models are summarized in Table 2.

To corroborate these findings, we repeated the competition reaction experiment, but this time using radio-labeled PPQ as substrate and cold CQ as inhibitor (Fig 4C). The experimental procedure and the data analysis were as described above. Again, partial noncompetitive inhibition and partial mixed-type inhibition best described the data, according to the Akaike information criterion difference ($\Delta AIC_C$) and the Akaike weight (Fig 4D). A subsequent discriminating F-test again favored the partial noncompetitive inhibition model ($p = 0.474$, F = 0.524, df = 37).

Noncompetitive and mixed-type inhibition occur when the inhibitor can bind to an allosteric site regardless of whether the substrate binding site is occupied or not [31]. The difference between the two models lies in the value of α, a parameter describing how much the affinity for the substrate changes when the inhibitor is already bound, and vice versa [31]. In the case of noncompetitive inhibition, α = 1, i.e., the affinity for the substrate does not change when the inhibitor is already bound. In comparison, in a mixed-type inhibition, α ≠ 1 and α > 0, i.e., the affinity changes. As a second approach to determine which of the two models best describes the competition between CQ and PPQ on PfCRT$^{Dd2}$, we fit the Michaelis-Menten

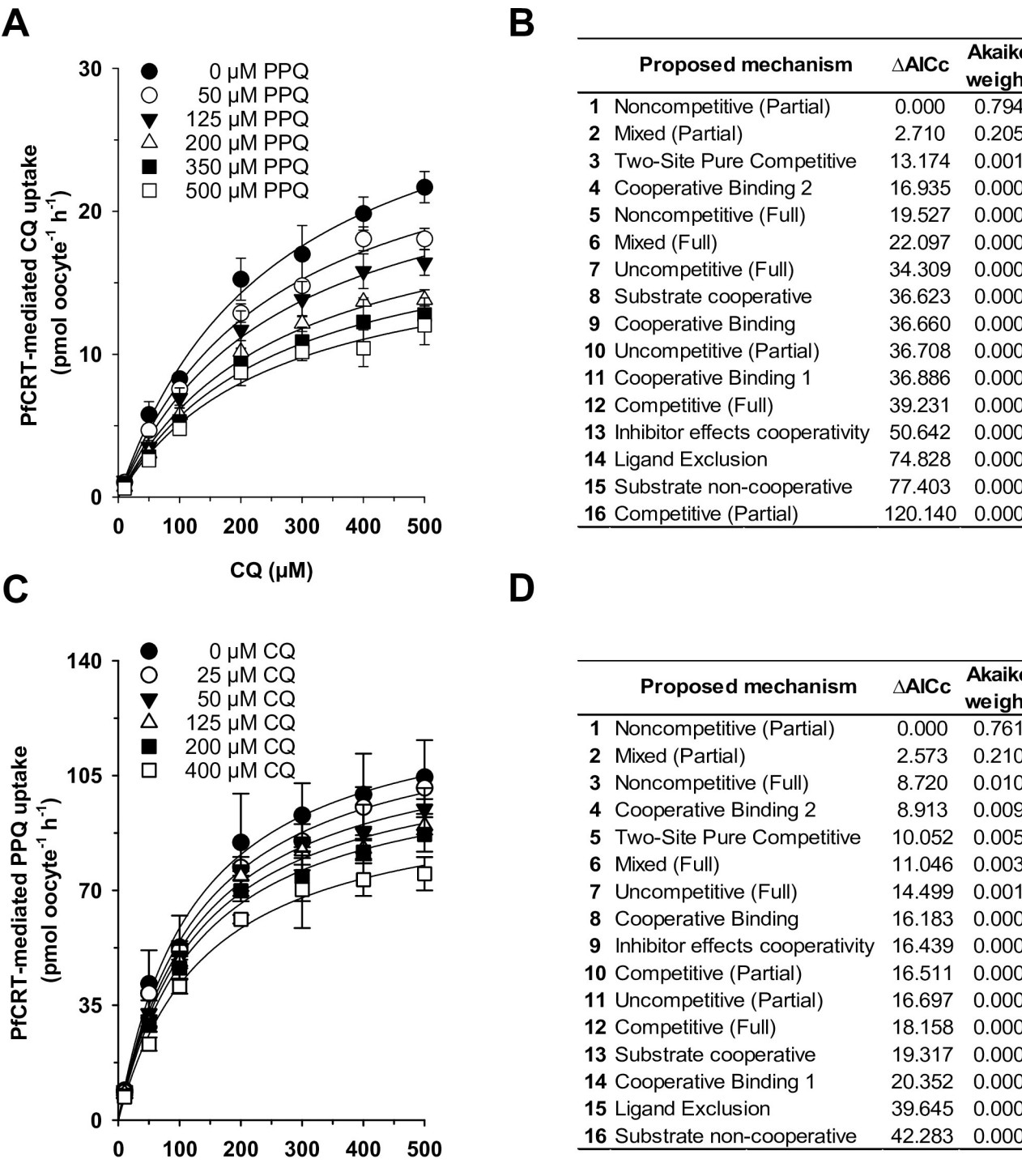

**Fig 4. CQ and PPQ are partial noncompetitive inhibitors of PfCRT<sup>Dd2</sup>-mediated drug transport.** A. PfCRT<sup>Dd2</sup>-mediated transport kinetics of radiolabeled CQ in the presence of increasing concentration of cold PPQ. B. Sixteen different models of enzymatic inhibition were globally fit to the data presented in (A), using the least-squares method. The likelihood of each model describing the data was evaluated by calculating the Akaike information criterion difference ($\Delta AIC_C$) and the Akaike weight, with the most plausible model ranking first. C, D. The same experiment as in (A), but this time using radiolabeled PPQ and cold CQ. The mean ± SEM of at least 3 biological replicates per condition are shown.

**Table 2. Kinetic parameters describing the interaction of PfCRT$^{Dd2}$ with CQ and PPQ.** Shown are the parameters from the two best models explaining the data presented in Fig 4. V$_{max}$, maximal velocity of drug transport; K$_S$, dissociation constant for the PfCRT-substrate complex; β, partiality factor; α, K$_S$ interaction factor; SEM, standard error; CI, confidence interval. N, number of determinations per kinetic analysis; n, total number of oocytes. For the CQ (PPQ) kinetics: N = 35, n = 838; for the PPQ (CQ) kinetics: N = 35, n = 545.

| Substrate (inhibitor) | Parameter | Noncompetitive (Partial) | | Mixed (Partial) | |
|---|---|---|---|---|---|
| | | Mean ± SEM | 95% CI | Mean ± SEM | 95% CI |
| CQ (PPQ) | V$_{max}$ (pmol oocyte$^{-1}$ h$^{-1}$) | 32 ± 1 | 30–34 | 33 ± 1 | 30–35 |
| | K$_S^{CQ}$ (μM) | 250 ± 20 | 224–284 | 260 ± 25 | 207–306 |
| | K$_S^{PPQ}$ (μM) | 200 ± 30 | 133–269 | 210 ± 50 | 114–296 |
| | α | | | 1.0 ± 0.2 | 0.49–1.44 |
| | β$^{CQ}$ | 0.38 ± 0.04 | 0.31–0.46 | 0.38 ± 0.06 | 0.27–0.49 |
| PPQ (CQ) | V$_{max}$ (pmol oocyte$^{-1}$ h$^{-1}$) | 131 ± 2 | 126–136 | 130 ± 3 | 123–136 |
| | K$_S^{PPQ}$ (μM) | 130 ± 6 | 118–142 | 125 ± 9 | 107–143 |
| | K$_S^{CQ}$ (μM) | 240 ± 80 | 75–402 | 210 ± 80 | 60–369 |
| | α | | | 1.2 ± 0.3 | 0.66–1.69 |
| | β$^{PPQ}$ | 0.60 ± 0.06 | 0.48–0.71 | 0.63 ± 0.08 | 0.48–0.78 |

equation to each of the six data sets from Fig 4A and 4C, yielding values for the apparent V$_{max}$ and the apparent K$_M$ for each condition. The resulting V$_{max,app}$ values and the V$_{max,app}$/K$_{M,app}$ ratios were plotted as a function of the inhibitor concentration (Fig 5A). The half-maximum inhibitory concentrations derived from the left panel in Fig 5A correspond to the dissociation constant between PPQ and the PfCRT$^{Dd2}$-CQ complex (αK$_S^{PPQ}$), which was 205 ± 50 μM, and to the dissociation constant between CQ and the PfCRT$^{Dd2}$-PPQ complex (αK$_S^{CQ}$), which was 220 ± 85 μM. Similarly, the half-maximum inhibitory concentrations derived from the right panel in Fig 5A correspond to the dissociation constant between PPQ and PfCRT$^{Dd2}$ (K$_S^{PPQ}$), which was 200 ± 40 μM, and to the dissociation constant between CQ and PfCRT$^{Dd2}$ (K$_S^{CQ}$), which was 250 ± 170 μM. Dividing αK$_S$ by the corresponding K$_S$ yielded values for α, which for both inhibition scenarios were not different from 1 (α = 1.0 ± 0.3 for prebound CQ and α = 0.9 ± 0.7 for prebound PPQ). This finding indicates that prebound CQ does not change the affinity of the PfCRT$^{Dd2}$ for PPQ and vice versa, consistent with a noncompetitive inhibition mechanism.

If CQ and PPQ are indeed partial noncompetitive inhibitors of PfCRT$^{Dd2}$, then depicting the kinetic data in the form of a fractional velocity plot should give a straight line that intercepts the y-axis at β(1- β), where β is the factor by which the CQ transport velocity is affected by PPQ [32]. In comparison, in cases of full noncompetitive inhibition, the straight line would pass through the origin [32]. The results obtained confirmed that the inhibition is indeed partial, and that PPQ slows down CQ transport via PfCRT$^{Dd2}$ by β$^{CQ}$ = 0.35 ± 0.06.

We next modelled the kinetics of the interaction between PfCRT$^{Dd2}$, CQ and PPQ, using the partial noncompetitive inhibition equation, and superimposed the modelled with the experimentally derived data in a 3D plot (Fig 5C and 5E). To assess how well the model fitted the experimental data, we calculated the difference between the observed and the predicted uptake values [33] and plotted the resulting residuals as a function of the CQ or the PPQ concentration (Fig 5D and 5F). The residuals were randomly distributed in all cases, thereby providing further evidence that partial noncompetitive inhibition is a plausible model for the interaction of PfCRT$^{Dd2}$ with CQ and PPQ.

A complementary set of experiments was undertaken to investigate the kinetic mechanism by which PfCRT$^{Dd2\_F145I}$ accepts CQ and PPQ (Fig 6). Again, all assays and tests favored the partial noncompetitive inhibition mechanism as the most plausible kinetic model to explain the interaction of CQ and PPQ on PfCRT$^{Dd2\_F145I}$. These included a global fit of the substrate

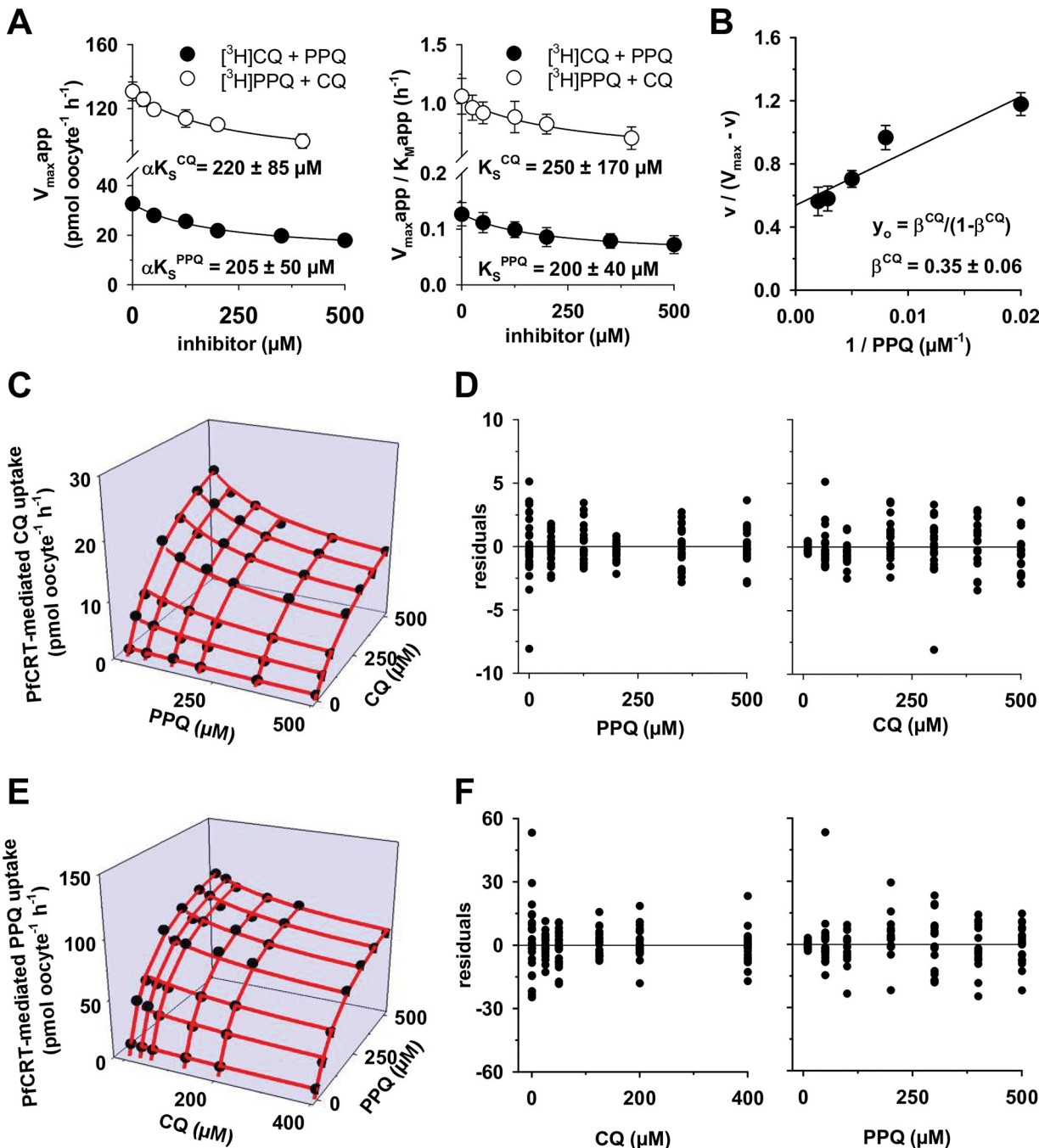

**Fig 5. Modeling the interaction of CQ and PPQ with PfCRT$^{Dd2}$ using a noncompetitive mechanism.** A. The apparent maximal velocity of transport ($V_{max}$app), as obtained by fitting the Michaelis-Menten equation to the data presented in Fig 4A and 4C, was analyzed as a function of the respective inhibitor concentration (*left panel*), yielding hyperboles whose half-maximal inhibitory concentration equated to $\alpha K_S^{CQ}$ (open circles) or to $\alpha K_S^{PPQ}$ (filled circles). *Right panel*, the ratios of $V_{max}$app to the corresponding apparent Michaelis constants ($K_M$app) were analyzed as a function of the respective inhibitor concentration, yielding hyperboles whose half-maximal inhibitory concentration equated to $K_S^{CQ}$ (open circles) or to $K_S^{PPQ}$ (filled circles). The value of $\alpha$ can be determined by dividing $\alpha K_S^{CQ}$ by $K_S^{CQ}$ or $\alpha K_S^{PPQ}$ by $K_S^{PPQ}$. The mean ± SD is shown per condition. B. The value of the partiality factor $\beta$ was derived by plotting the ratio of uptake (v) to $V_{max}$—v versus the reciprocal of the inhibitor concentration [32]. C. The inhibition of CQ transport by PPQ was modelled using the partial noncompetitive equation and displayed as a 3D plot (red lines). The following parameters were used: $V_{max}^{CQ}$ = 32 pmol oocyte$^{-1}$ h$^{-1}$; $K_S^{CQ}$ = 254 μM; $K_S^{PPQ}$ = 201 μM; $\beta^{CQ}$ = 0.38. The experimentally derived data were overlaid for comparison (black circles). D. The difference between experimentally derived uptake values and the predicted values was calculated, and the resulting residuals were plotted as a function of the PPQ concentration (*left*) or the CQ concentration (*right*) [33]. E. as in (C), but inhibition of PPQ transport by CQ. The following parameters were used: $V_{max}^{PPQ}$ = 131 pmol oocyte$^{-1}$ h$^{-1}$; $K_S^{CQ}$ = 238 μM; $K_S^{PPQ}$ = 130 μM; $\beta^{PPQ}$ = 0.60. F. as in (D), but for the inhibition of PPQ transport by CQ.

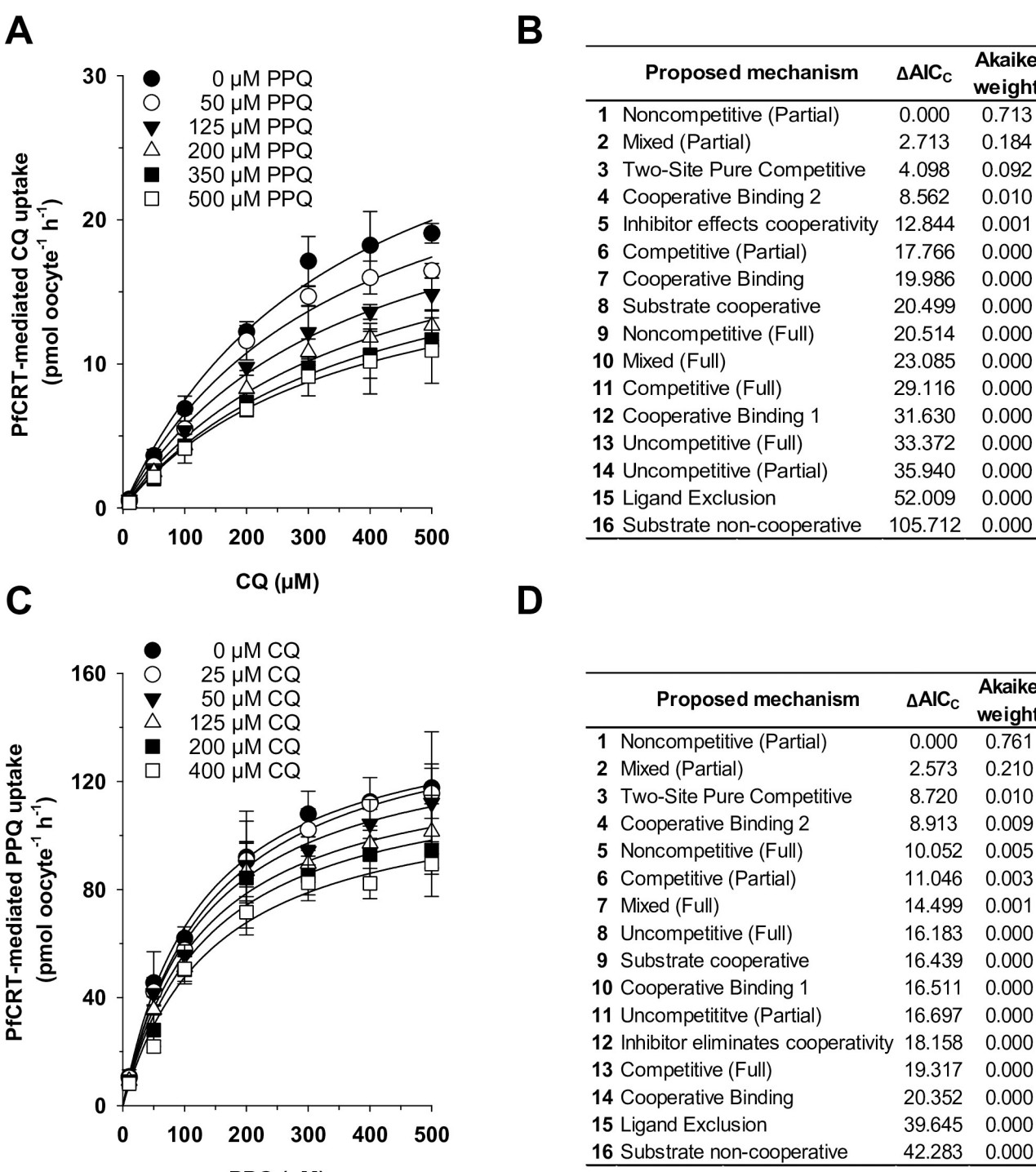

**Fig 6. CQ and PPQ are partial noncompetitive inhibitors of PfCRT^Dd2_F145I-mediated drug transport.** A. PfCRT^Dd2_F145I-mediated transport kinetics of radiolabeled CQ in the presence of increasing concentrations of cold PPQ. B. Sixteen different models of enzymatic inhibition were globally fit to the data presented in (A), using the least-squares method. The likelihood of each model describing the data was evaluated by calculating the Akaike information criterion difference ($\Delta AIC_C$) and the Akaike weight, with the most plausible model ranking first. C, D. The same experiment as in (A), but this time using radiolabeled PPQ and cold CQ. The mean ± SEM of at least 3 biological replicates per condition are shown.

competition data to the 16 kinetic models (Fig 6), analysis of the secondary plots ($V_{max,app}$ and $V_{max,app}/K_{M,app}$ as a function of the inhibitor concentration) (Fig 7A), a fractional velocity plot analysis (Fig 7B) and a comparison of the experimental data with the modelled data using the noncompetitive inhibition equation (Fig 7C to 7F). The secondary plots confirmed that the factor α is statistically not different from 1 (α = 1.0 ± 0.3 for prebound CQ and α = 0.8 ± 0.7 for prebound PPQ), indicating that the affinity of PfCRT$^{Dd2\text{–}F145I}$ for either of the two drugs is not altered when one or the other drug is already bound. The fractional velocity plot illustrated the partial character of the noncompetitive inhibition mechanism by yielding a partiality factor β$^{CQ}$ = 0.31 ± 0.04, by which PPQ reduces the transport velocity of CQ (Fig 7B). Thus, PfCRT$^{Dd2\text{–}F145I}$, like PfCRT$^{Dd2}$, can simultaneously accept both CQ and PPQ at independent binding sites without changes in affinity in a partial noncompetitive inhibition mechanism. A 3D model of the kinetics between PfCRT$^{Dd2\text{–}F145I}$, CQ and PPQ, and analyses of the residuals as a function of substrate and inhibitor concentrations confirmed that the model was also an appropriate explanation for the mutant transporter (Fig 7C to 7F). The parameters from the fit to the partial noncompetitive and the partial mixed-type inhibition models are summarized in Table 3.

## MD simulations support allosteric binding model

To further test the allosteric binding mechanism, we modeled the tertiary structure of PfCRT$^{Dd2}$ by homology using the cryo-EM structure of the related CQ-resistance conferring PfCRT$^{7G8}$ isoform (PDB ID: 6UKJ; 3.2 Å) [9] as a template. The latter is in an open-to-vacuole conformation and it is in complex with a positively charged Fab fragment that is accommodated in the negative transporter cavity on the digestive vacuole side and was not retained in the modeling template. The high sequence identity between the two isoforms (98%) enabled the construction of a high-quality model as verified using the MolProbity utility implemented in SWISS-MODEL [34] (S4 Fig). The binding of the two substrates to the transporter was then investigated by molecular docking. Considering the physiological pH of the digestive vacuole of 5.2 [35], the substrates were modeled in their protonated forms (CQ$^{2+}$ and PPQ$^{4+}$). Because the template structure contained an antibody fragment that was absent in the model, induced-fit docking (IFD) was performed. The whole of the transporter cavity was considered for docking and up to ten poses were generated for the two compounds, as described in the Materials and Methods.

The docking results reflect the differences in the experimentally measured affinities of the two compounds for PfCRT$^{Dd2}$, with CQ having less favorable docking scores compared to PPQ (S1 Table). The generated poses show that CQ can be accommodated in two main regions: the top central part and the left side of the channel (S5 Fig). The latter, defined by TM helices 1, 2, 3 and 7, is occupied by the majority of the poses (ranked 3 to 10) and agrees with the experiments suggesting CQ interacts with F145. The best-scored of these poses (pose 3 in S1 Table, docking score of -5.734 kcal/mol) is shown in Fig 8A. In this pose, CQ forms polar contacts via the two positively charged nitrogens: the nitrogen of the diethylamino tail makes a salt bridge with E75 on TM1 and the nitrogen of the 4-amminoquinoline ring makes a hydrogen bond with S257 on TM7. Hydrophobic contacts are established by the quinoline ring with I260, V141, and F145. Substituting alanine for each of V141, S257, I260 significantly reduced PfCRT-mediated CQ transport in all three cases, as compared with PfCRT$^{Dd2}$ (p<0.001; ANOVA test) (S6 Fig).

PPQ shows greater variability in the generated poses, with the first two best ranked poses being located on the right side of the channel and the others displaying a bending towards a horizontal that fully occupies the cavity (S5 Fig), with poses 6 to 9 (S1 Table) traversing the

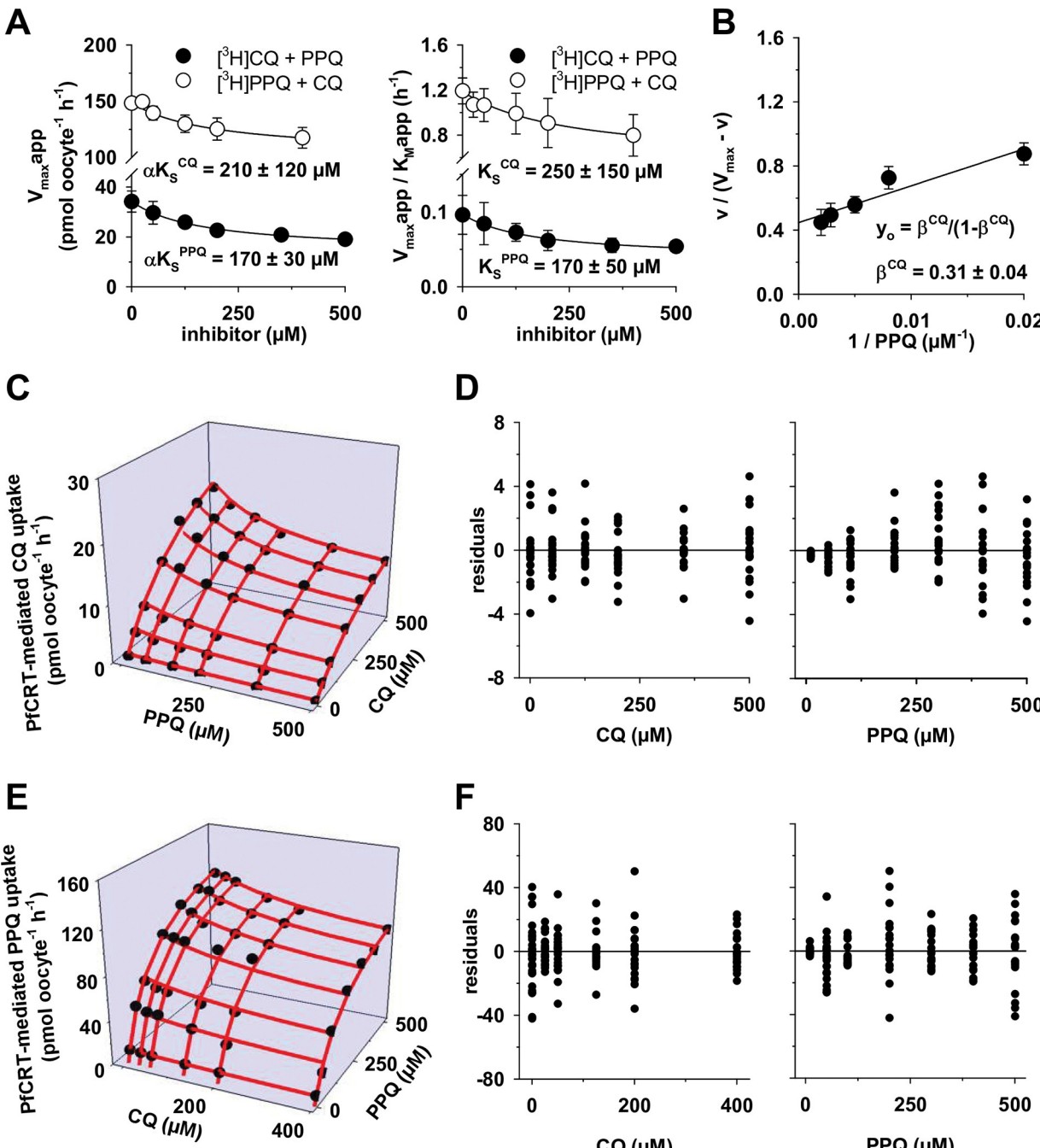

**Fig 7. Modeling the interaction of CQ and PPQ with PfCRT$^{Dd2\_F145I}$ using a noncompetitive mechanism.** A. The apparent maximal velocity of transport ($V_{max}$app), as obtained by fitting the Michaelis-Menten equation to the data presented in Fig 6A and 6C, was analyzed as a function of the respective inhibitor concentration (*left panel*), yielding hyperboles whose half-maximal inhibitory concentration equated to $\alpha K_S^{CQ}$ (open circles) or to $\alpha K_S^{PPQ}$ (filled circles). *Right panel*, the ratios of $V_{max}$app to the corresponding apparent Michaelis constants ($K_M$app) were analyzed as a function of the respective inhibitor concentration, yielding hyperboles whose half-maximal inhibitory concentration equated to $K_S^{CQ}$ (open circles) or to $K_S^{PPQ}$ (filled circles). The value of $\alpha$ can be determined by dividing $\alpha K_S^{CQ}$ by $K_S^{CQ}$ or $\alpha K_S^{PPQ}$ by $K_S^{PPQ}$. The mean ± SD is shown per condition. B. The value of the partiality factor $\beta$ was derived by plotting the ratio of uptake (v) to $V_{max}$—v versus the reciprocal of inhibitor concentration [32]. C. The inhibition of CQ transport by PPQ was modelled using the partial noncompetitive equation and displayed as a 3D plot (red lines). The following parameters were used: $V_{max}^{CQ}$ = 34 pmol oocyte$^{-1}$ h$^{-1}$; $K_S^{CQ}$ = 357 µM; $K_S^{PPQ}$ = 167 µM; $\beta^{CQ}$ = 0.40. The experimentally derived data were overlaid for comparison (black circles). D. The difference between experimentally derived uptake values and the predicted values was calculated, and the resulting residuals were plotted as a function of the PPQ concentration (*left*) or the CQ concentration (*right*) [33]. E. as in (C), but inhibition of PPQ transport by CQ. The following parameters were used $V_{max}^{PPQ}$ = 152 pmol oocyte$^{-1}$ h$^{-1}$; $K_S^{CQ}$ = 225 µM; $K_S^{PPQ}$ = 134 µM; $\beta^{PPQ}$ = 0.61. F. as in (D), but for the inhibition of PPQ transport by CQ.

**Table 3. Kinetic parameters describing the interaction of PfCRT$^{Dd2\_F145I}$ with CQ and PPQ.** Shown are the parameters from the two best models explaining the data presented in Fig 6. $V_{max}$, maximal velocity of drug transport; $K_S$, dissociation constant for the PfCRT-substrate complex; β, partiality factor; α, $K_S$ interaction factor; SEM, standard error; CI, confidence interval. N, number of determinations per kinetic analysis; n, total number of oocytes. For the CQ (PPQ) kinetics: N = 31, n = 630; for the PPQ (CQ) kinetics: N = 41, n = 625.

| Substrate (inhibitor) | Parameter | Noncompetitive (Partial) | | Mixed (Partial) | |
|---|---|---|---|---|---|
| | | Mean ± SEM | 95% CI | Mean ± SEM | 95% CI |
| CQ (PPQ) | $V_{max}$ (pmol oocyte$^{-1}$ h$^{-1}$) | 34 ± 2 | 31–38 | 34 ± 2 | 29–39 |
| | $K_S^{CQ}$ (μM) | 360 ± 30 | 299–416 | 350 ± 50 | 260–445 |
| | $K_S^{PPQ}$ (μM) | 170 ± 30 | 103–231 | 160 ± 40 | 83–244 |
| | α | | | 1.0 ± 0.3 | 0.39–1.69 |
| | $β^{CQ}$ | 0.40 ± 0.04 | 0.32–0.47 | 0.41 ± 0.08 | 0.25–0.56 |
| PPQ (CQ) | $V_{max}$ (pmol oocyte$^{-1}$ h$^{-1}$) | 152 ± 4 | 145–160 | 150 ± 5 | 141–160 |
| | $K_S^{PPQ}$ (μM) | 134 ± 8 | 119–150 | 128 ± 11 | 105–151 |
| | $K_S^{CQ}$ (μM) | 230 ± 90 | 39–411 | 200 ± 90 | 27–374 |
| | α | | | 1.2 ± 0.3 | 0.56–1.82 |
| | $β^{PPQ}$ | 0.61 ± 0.07 | 0.47–0.74 | 0.64 ± 0.09 | 0.47–0.82 |

channel. In the top-ranked pose (-9.329 kcal/mol), PPQ occupies the right side of the cavity, surrounded by TM1, 4, 6, 8, and 9 (Fig 8B). In this position, the protonated nitrogens of the two piperazine moieties establish hydrogen bonds with D329 on TM8 and T356 on TM9 while the quinoline ring pointing at TM8 makes a π-π interaction with Y345 on TM9. The other quinoline ring is positioned between TM9 and TM8 on the digestive vacuole side. Hydrophobic contacts are formed by the two piperazine moieties with V224 and L160.

To check for their stability, the two docking poses described above were further investigated by MD simulations. In accordance with both the low affinity and poor docking score, CQ is quite mobile within the binding site since from the very beginning of the simulation and it significantly changes its location with respect to its original position (S7 Fig). Already within the first 100 ns of simulation, CQ leaves the hydrophobic region around F145 to gradually move to the top-right part of the cavity, already explored by docking. During this transition, CQ continuously rearranges in the central part of the cavity while maintaining contact with E75, finally visiting the right side of the channel towards the end of the simulation (S8 Fig). PPQ also experiences a large conformational rearrangement to an arrangement that is maintained for most of the trajectory length (~150–1000 ns) (S7 Fig). In this pose, PPQ is pulled down to the digestive vacuole side with one aminoquinoline ring anchored to E75 and the other pointing backward to the loop connecting JM2 and TM7. Still preserving the bond with E75, PPQ readjusts the quinoline ring in the last part of the trajectory to make a hydrogen bond to E198, assuming an orientation that is maintained until the end of the simulation (S8 Fig).

MD simulations were also employed to simulate the coexistence of CQ and PPQ within the transporter and to further support the noncompetitive nature of the two drugs, as observed experimentally. CQ and PPQ were simulated together within PfCRT$^{Dd2}$ after merging the original docking coordinates illustrated in Fig 8, *i.e.*, pose 3 for CQ and pose 1 for PPQ. As a result of the presence of both drugs (Fig 9), CQ only fluctuates on the left side of the channel without moving to the right part and experiences two main arrangements (S7 Fig). One is adopted during the central part of the simulation (300–600 ns) in which CQ moves downwards to the hydrophobic region around F145 (Fig 9C). The other, more stable arrangement has CQ occupying the upper part of the cavity, making hydrogen bonds with D137 and N98 and hydrophobic contacts with F101, V141, and F322 and being stable to the end of the simulation (Fig 9D). PPQ undergoes a large conformational adjustment (~15 Å) which pulls down the compound to the digestive vacuole to a greater extent than when PPQ is simulated alone

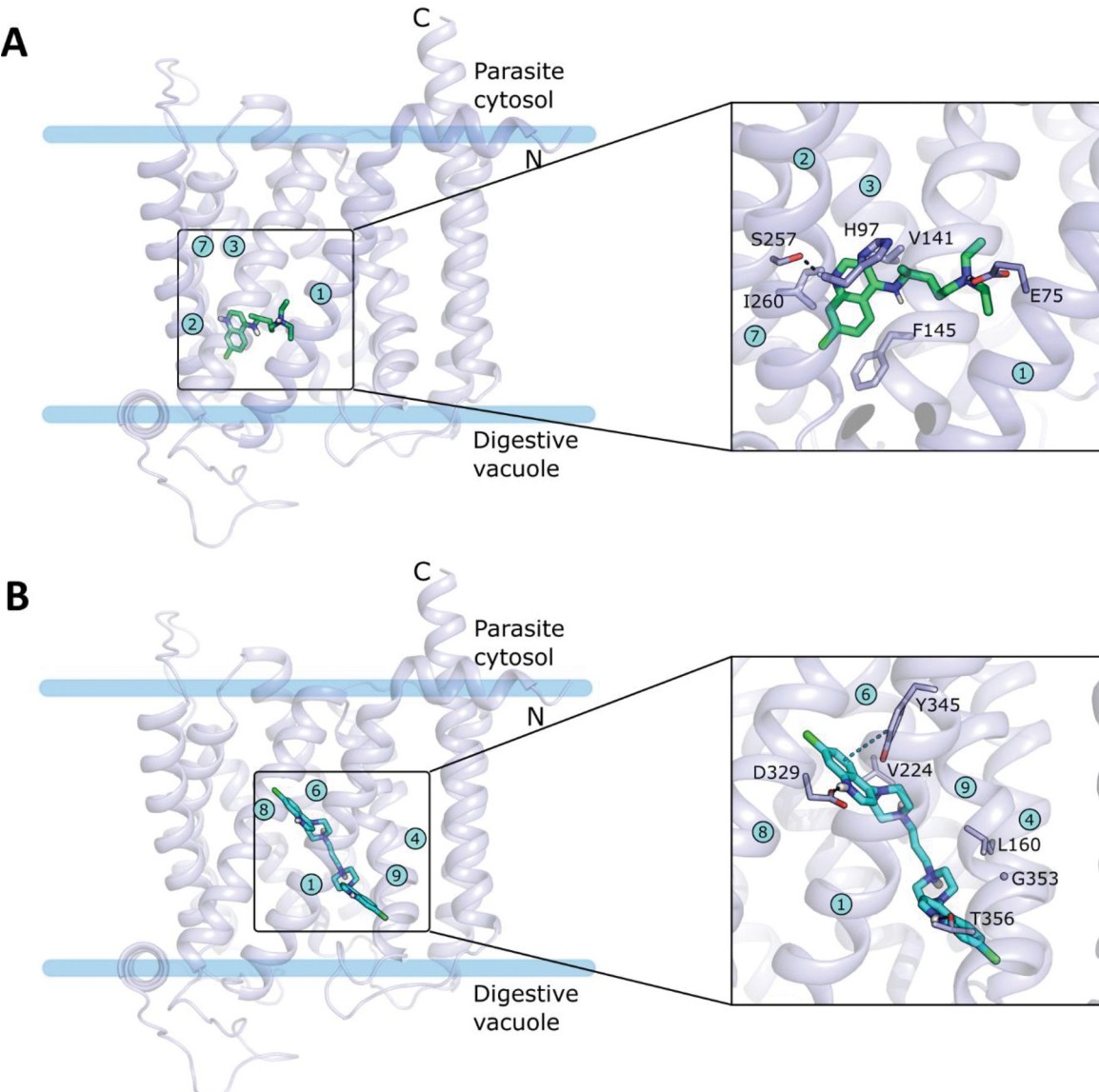

**Fig 8.** Docking poses of CQ (A) and PPQ (B) in the cavity of PfCRT[Dd2]. The docking poses (pose 3 for CQ and pose 1 for PPQ) within the whole protein structure are shown with close-ups of the binding pocket and the established interactions in the insets. The protein is shown in transparent light blue cartoon representation, and the substrates and the interacting residues are shown as sticks. The N- and C- termini, as well as the transmembrane helices (blue circles with numbers) and the residues are labeled. Interactions between ligands and the protein are shown by dashed lines: hydrogen bonds and salt-bridges in black and π-π interactions in grey.

(S7 Fig). This arrangement is later stabilized by a methionine-aromatic interaction between M305 of the vacuolar loop and the quinoline ring, and by a polar contact between the protonated piperazine and E207. Additional hydrophobic interactions are made by the quinoline and piperazine rings with T205, V369, T356, and L160 (Fig 9B). Interestingly, the vacuolar loop is key to firmly anchoring PPQ uniquely when the two compounds are simulated together, while it behaves differently in the other systems indicating that it plays a role in the stabilization of the complexes. A comparison of the loop motion is depicted in S9 Fig and shows its conformational rearrangements over the simulation time. When CQ is simulated alone with PfCRT[Dd2], an open conformation is adopted by the loop at the beginning of the simulation, and

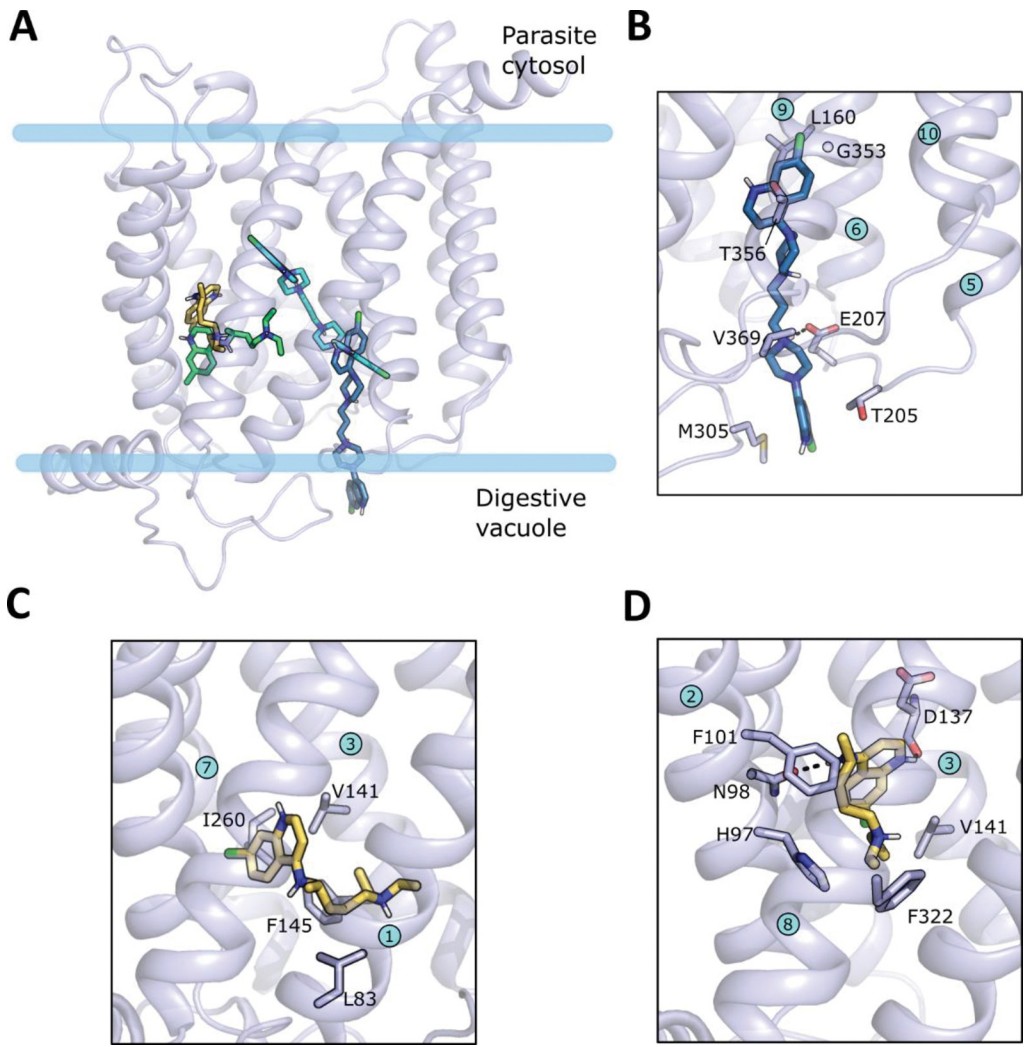

**Fig 9. MD simulation of PfCRT[Dd2] in complex with both CQ and PPQ.** A. The last frame from the simulation is shown and superimposed on the original docking poses of CQ and PPQ. B. Interactions established by PPQ during the last hundreds of ns. C. Pose and interactions of CQ in the central part of the simulation (~400–600 ns). D. Final pose and interactions established by CQ in the last hundreds of ns. The docking and the MD final poses are shown in green and yellow, respectively, for CQ, and cyan and blue, respectively, for PPQ. The protein and the amino acid residues that interact with ligands are shown in light blue.

correspondingly CQ gradually moves to the right side of the channel, and it is then stably maintained there to the end. The same final open conformation is reached by the loop when PPQ is simulated alone but it is accompanied by increased fluctuation over time. Conversely, a progressive closure of the loop is observed when the two compounds are simulated together. In this case, while initially assuming an open conformation similar to the one described above, the loop moves inwards towards the cavity when PPQ moves down towards the digestive vacuole to an arrangement that locks access to the channel. In a previous study, it was proposed that PPQ and CQ act as competitive inhibitors of each other on PfCRT[7G8]. To explore a possible isoform-specific binding behavior, we conducted a docking analysis using PfCRT[7G8]. While the docking of CQ to PfCRT[7G8] agreed with that of PfCRT[Dd2], PPQ mainly occupied the center-left side of the cavity without entirely occupying the right side (Fig 10A–10E). The amino acid sequence differences on the right side of the transporter (I356 and R371 in

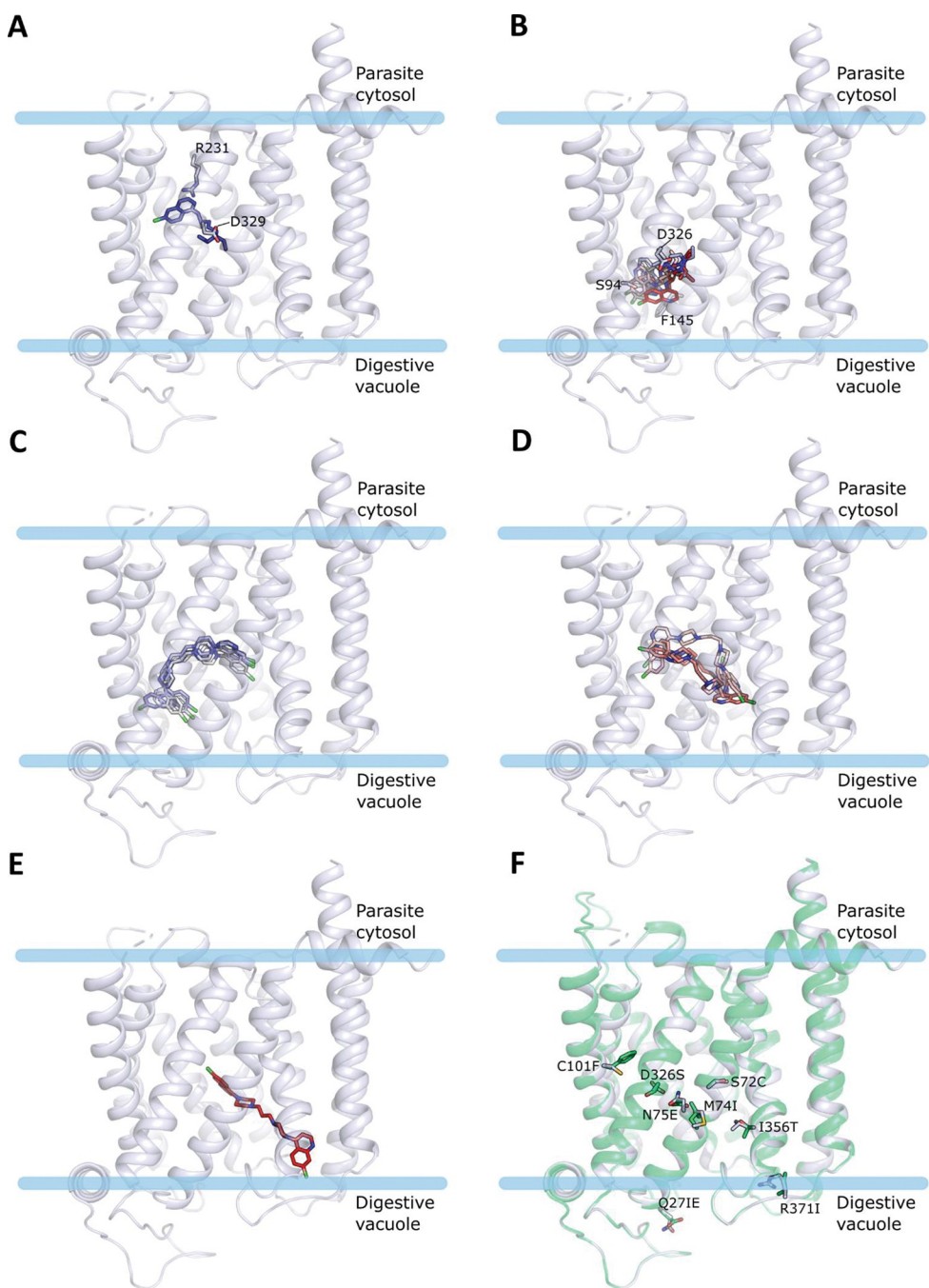

**Fig 10. Most representative docking results of CQ and PPQ to PfCRT^7G8.** A blue-to-red color scale shows the generated docking poses for CQ (**A-B**) and PPQ (**C-E**), ranked from the best (blue) to the worst (red) docking score. **F.** Superposition of the Dd2 (green) and the 7G8 (white) isoform with the different residues shown as sticks and labeled.

PfCRT^7G8 and T356 and I371 in PfCRT^Dd2) (Fig 10F) may impede PPQ accommodation in the 7G8 isoform, leading to overlapping binding sites for CQ and PPQ. These results support the suggested competitive binding of the drugs to PfCRT^7G8 [9], and suggest a noncompetitive binding model specific to the PfCRT^Dd2 isoform.

## Docking simulations to the PPQ-resistant PfCRT variants

Docking to the other tested PPQ-resistant mutants, PfCRT$^{Dd2\_H97Y}$, PfCRT$^{Dd2\_F145I}$, PfCRT$^{Dd2\_G353V}$ and PfCRT$^{Dd2\_M343L}$, showed both drugs bind similarly to PfCRT$^{Dd2}$ but did not provide a clear distinction between the different mutants in terms of docking scores with all being within 1–2 kcal/mol of the respective score for PfCRT$^{Dd2}$ for each compound (S1 Table). In the F145I mutant, the absence of the aromatic residue results in the 3 top-ranked poses of CQ being positioned in the upper left side of the channel (making interactions with D329 and R231), and a second cluster of poses is in the vicinity of position 145 (predominantly making interactions with H97, F322, D329 and/or E75) (S10A and S10B Fig). PPQ mainly docks on the right side of the channel, with some poses in agreement with the top-ranked ones found in PfCRT$^{Dd2}$, further supporting the possibility of PfCRT hosting both CQ and PPQ in two distinct sites (S10C–S10F Fig), as observed in the kinetics experiment. The H97Y mutation results in CQ being mostly confined to the bottom left side of the channel in the neighborhood of F145 where key contacts are established with F145, F322, and Y97, highlighting the importance of aromatic interactions for CQ binding, and with E75 (S11A and S11B Fig). PPQ instead shows a variegated profile in which the majority of the poses fill the whole cavity without a predominant orientation (S11C–S11E Fig). When G353 is mutated to valine, CQ remains mainly positioned around F145, alternating interactions with H97, F322, F145, E75, and S94, while PPQ adopts different orientations, mostly fully occupying the cavity (S12 Fig). However, the replacement of the glycine with a bulky residue, such as the valine in question, is likely expected to perturb the structure of TM2 and result in structural adjustments of the neighboring regions that may not be fully captured by the docking procedure and that will differently affect the PPQ binding poses. Docking to the M343L mutant does not show many differences to the PfCRT$^{Dd2}$ as M343 is located at the top of TM9 pointing to TM4-5, does not line the central cavity, and is outside the box used for defining the docking grid (S13 Fig). Similar to G353V, the M343L mutation effects are expected to be due to larger structural rearrangements that cannot be accounted for solely via docking.

## Non-Michaelis Menten kinetics for an engineered PfCRT variant

Given that the molecular docking and the kinetic analyses suggest a role for F145 in the binding of CQ, we wondered whether the effect of the F145I replacement in reducing the affinity for CQ could be compensated by combining this mutation with the H97Y replacement, the latter increasing the affinity for CQ in the PfCRT$^{Dd2}$ background (Table 1). For this purpose, we generated the respective PfCRT$^{Dd2\_H97Y\_F145I}$ double mutant (for expression levels see S1 and S2 Figs) and examined its kinetic properties in the *X. laevis* oocyte system, as a function of increasing CQ and PPQ concentrations (Fig 11). Unexpectedly, fitting the Michaelis-Menten equation to the resulting data points yielded a result with low confidence for both CQ and PPQ kinetics. The confidence improved significantly when using a Hill equation, as corroborated by F-statistics (CQ, $p = 0.043$, df = 4, F = 10.631; PPQ, $p = 0.037$, df = 4, F = 11.905). According to the kinetic parameters retrieved from the Hill equation, PfCRT$^{Dd2\_H97Y\_F145I}$ displayed a $V_{max}$ for CQ of $35 \pm 1$ pmol oocyte$^{-1}$ h$^{-1}$, which is comparable to that of PfCRT$^{Dd2}$ and PfCRT$^{Dd2\_F145I}$. In comparison, the $K_M$ for CQ ($240 \pm 40$ µM) decreased in relation to that of PfCRT$^{Dd2\_F145I}$ but increased from that of PfCRT$^{Dd2\_H97Y}$, and attained a value comparable to that PfCRT$^{Dd2}$. As to PPQ, the double mutation, H97Y and F145I, did not significantly affect the $V_{max}$ for PPQ ($131 \pm 10$ pmol oocyte$^{-1}$ h$^{-1}$), and instead increased the corresponding $K_M$ value to $170 \pm 20$ µM, suggesting that the carrier has a lower affinity for PPQ as compared with PfCRT$^{Dd2}$ or the single mutation variants. The Hill coefficients, $H_c$, were $1.6 \pm 0.2$ for CQ and $2.0 \pm 0.3$ for PPQ, indicative of a carrier that can accept and transport two molecules of

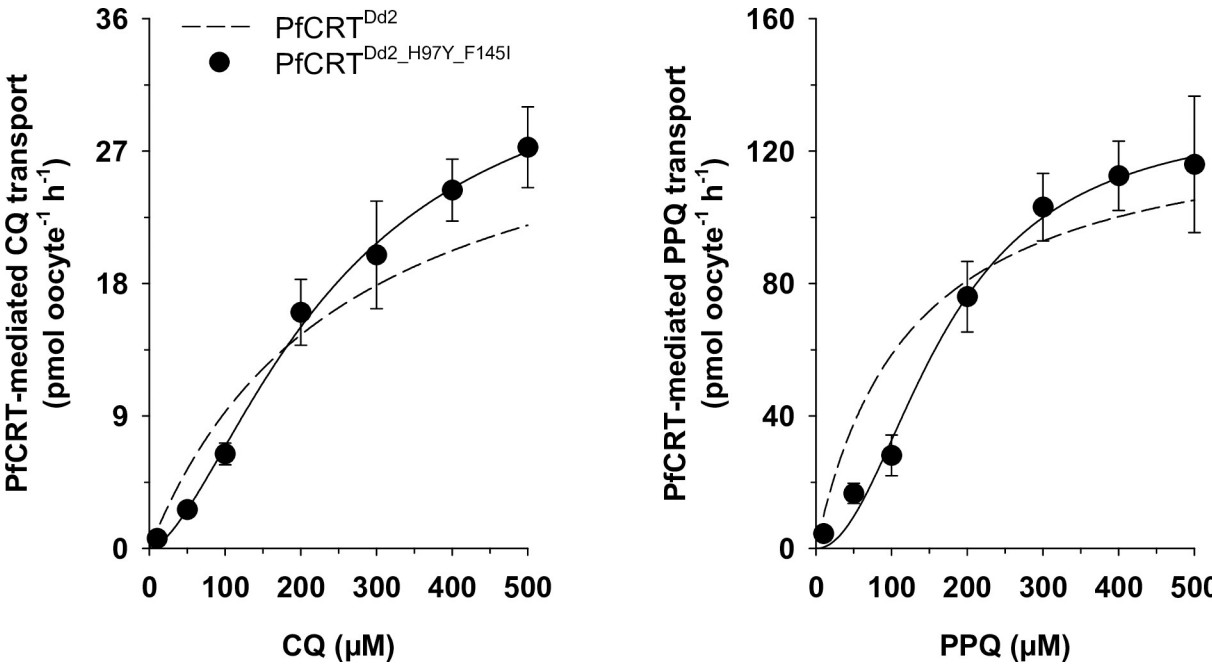

**Fig 11. Non-Michaelis Menten kinetics for PfCRT[Dd2_H97Y_F145I] combining two PPQ resistance-conferring mutations.** The PfCRT[Dd2_H97Y_F145I]-mediated uptake of CQ (*left panel*) and PPQ (*right panel*) was measured over an extracellular concentration range of 10 to 500 μM (radiolabeled plus unlabeled), at pH 6. A Hill function was fit to the data points, yielding the kinetic parameters presented in Table 1. The mean ± SEM are shown of at least 3 independent biological replicates per condition. For comparison, the kinetics of PfCRT[Dd2] is shown as a dashed line. The goodness of fit was compared between a Michaelis-Menten function and a Hill function and in each case a Hill function was statistically favored, according to F-statistics (CQ, $p = 0.043$, df = 4, F = 10.631; PPQ, $p = 0.037$, df = 4, F = 11.905).

CQ or PPQ. Noteworthy, in the range between 10 and 200 μM CQ or PPQ, PfCRT[Dd2_H97Y_F145I] transported less drug per time unit than did PfCRT[Dd2] or any of the parental single mutants. However, at higher substrate concentrations, PfCRT[Dd2_H97Y_F145I] transported significantly more CQ per time unit than any other variant, and as much PPQ as the highly PPQ-transporting PfCRT[Dd2_F145I] isoform.

## Discussion

PPQ resistance is a complex phenotype in the parasite, and although PfCRT is a contributing factor, it is not the only one [8,27,36,37]. The PfCRT variants associated with PPQ resistance often incur a fitness cost due to their altered ability to handle oligopeptides [8,10,26], which are the natural substrate of PfCRT [17,18]. The focus of this study was solely on PfCRT and how mutational changes associated with PPQ resistance affect CQ and PPQ transport kinetics, with the aim of providing a mechanistic understanding of the CQ re-sensitizing phenomenon associated with PfCRT variants contributing to PPQ resistance. Although we took a reductionistic approach by focusing on PfCRT, we believe that our study's conclusions are not limited by this choice.

The CQ resistance transporter from the Southeast Asian *P. falciparum* strain Dd2 can handle various quinoline and quinoline-like antimalarial drugs [5,21,22,25]. This includes PPQ, a drug deployed in combination with dihydroartemisinin, as shown in some but not all studies [8–10,38]. However, this demonstrated PPQ transport activity is insufficient to confer clinical PPQ resistance. To mediate clinical PPQ resistance, PfCRT[Dd2] has to acquire an additional mutation, which can occur in different parts of the carrier [8]. Our study of four different PPQ resistance-associated PfCRT variants now shows that this additional mutation results in subtle

changes in the apparent PPQ $K_M$ and/or $V_{max}$ values. By increasing the affinity and/or the $V_{max}$ for PPQ, the additional mutation apparently enhances the transport efficiency of the carrier above a resistance conferring threshold. Minimal kinetic requirements for an optimized PPQ carrier would include a 10% or higher increase in $V_{max}$ and/or a 20% or higher decrease in the $K_M$, as compared with PfCRT$^{Dd2}$. The increased transport efficiency for PPQ comes at the expense of a reduced transport efficiency for CQ, with the $V_{max}$ value for PPQ anti-correlating with the $K_M$ value for CQ (Fig 2C). This anti-correlation in the kinetic transport parameters provides a plausible mechanistic explanation for the observed re-sensitization to CQ induced by the additional PPQ resistance conferring mutation.

Both CQ and PPQ share the 4-chloroquinoline moiety, with PPQ consisting of two 4-chloroquinoline groups connected via a di-piperazine propane bridge. On the basis of the structural similarities and considering the large size of PPQ, it has been proposed that PPQ and CQ occupy the same or at least overlapping binding sites on PfCRT [9]. Initial proximity-based binding assays, using PfCRT$^{7G8}$, supported this hypothesis by suggesting that PPQ and CQ are competitive inhibitors of one another [9]. Our kinetic analyses using the related PfCRT$^{Dd2}$ paint a more nuanced picture. Firstly, the competition plot experiment, which explicitly tests for competitive reactions, favored an allosteric binding model (Fig 3). Secondly, transport kinetics using either radiolabeled CQ or PPQ and the other drug in unlabeled form, together with model discriminating information theory and graphical secondary analyses (Figs 4, 5, 6 and 7), indicate that CQ and PPQ act as partial noncompetitive inhibitors of one another and not as competitive inhibitors. This was demonstrated for both PfCRT$^{Dd2}$ and PfCRT$^{Dd2\_F145I}$. Noncompetitive inhibition occurs when the substrate and the inhibitor bind at different sites on the enzyme or transporter, and the binding of either one does not affect binding of the other [31]. In the case of PfCRT$^{Dd2}$ and PfCRT$^{Dd2\_F145I}$, this means that CQ and PPQ can simultaneously bind with unaltered affinity, yielding a ternary complex that is capable of completing the transport cycle, yet with reduced transport rate, as indicated by a factor $\beta < 1$. Whether both drugs or only one drug is released at the end of the transport cycle is unclear and needs to be clarified in further studies. Both types of substrate release model can be found in various cases of noncompetitive inhibition studied in other systems [39–41].

The second most plausible model describing the interaction of PPQ and CQ with PfCRT$^{Dd2}$ was partial mixed-type inhibition. Mixed-type inhibition is similar to noncompetitive inhibition, with the exception that binding of one substrate reduces the affinity of the other substrate by a factor $\alpha$. In the case of both PfCRT$^{Dd2}$ and PfCRT$^{Dd2\_F145I}$, $\alpha$ was not different from 1 (between $1.0 \pm 0.2$ and $1.2 \pm 0.3$, depending on which of the two drugs was radiolabeled), which made the mixed-type inhibition model appear less likely than the noncompetitive inhibition model.

Docking and MD simulations, in which either CQ or PPQ or both drugs were docked into the modelled open-to-vacuole structure of PfCRT$^{Dd2}$, provided further support for the putative existence of two distinct binding sites. Interestingly, a comparison with the dockings of both drugs to PfCRT$^{7G8}$ suggests an isoform-specific behavior. While the docking of CQ to PfCRT$^{7G8}$ is pretty much in agreement with that to PfCRT$^{Dd2}$, PPQ mainly binds to the center-left side of the cavity with no poses entirely occupying the right side (Fig 10A–10E). The differences in the amino acid sequence of the right side of the transporter (I356 and R371 in PfCRT$^{7G8}$ and T356 and I371 in PfCRT$^{Dd2}$) (Fig 10F) could hamper the accommodation of PPQ in the PfCRT$^{7G8}$ isoform, resulting in overlapping binding sites for CQ and PPQ. These findings are in agreement with the proposed competitive binding of the two drugs to PfCRT$^{7G8}$ [9] and let us hypothesize that a noncompetitive binding model is specific for the PfCRT$^{Dd2}$ isoform. It seems to be a recurring theme that the substrate binding cavity of PfCRT is highly flexible and can offer alternative solutions to a particular drug challenge.

In PfCRT[Dd2], PPQ preferably occupied the right side of the channel where it was stably maintained during the simulation, after some structural adjustments. PPQ is further predicted to stay in the close vicinity to G353, whose mutation to valine confers higher affinity, both in the docking and simulations. Although hypothesizing that the G353V mutation could provide additional hydrophobic interaction with the quinoline ring, as an explanation for the increased affinity, we could not clearly observe this when docking PPQ to this mutant. However, the location of the G353V mutation is likely to impact conformational changes of the cavity that could differently affect PPQ binding, and this is also expected for the M343L mutant. Indeed, changes in the shape of the cavity are predicted to be determinants for conferring PPQ resistance [9,10]. In comparison, docking to PfCRT[Dd2] predicts CQ as having a more favorable interaction in the top-central part of the channel but still showing an entropically favored binding site in the hydrophobic region surrounded by F145 and H97. The latter region was also found to be the predominant binding site in the tested mutants, with conserved hydrophobic interactions (with F145, H97 and F322) that stress the importance of aromatic side chains for the binding of CQ. Overall, while mostly supporting a noncompetitive binding model, in agreement with the experimental evidence for PfCRT[Dd2] and PfCRT[Dd2_F145I], docking of the different mutants did not allow mutations that affect the drugs' affinities to be distinguished based on the docking score. As already noted, we believe conformational changes derived from a single mutation play a key role, and that accounting for flexibility via investigation of the dynamics of the system is pivotal for a deeper understanding.

MD simulations show the propensity of CQ to bind to the upper-central part of the channel when simulated alone, in agreement with the docking results. However, CQ has a low affinity for PfCRT[Dd2], as also reflected in the poor docking scores, which indicate a less strong interaction within the transporter and result in high instability and fluctuation within the binding site during simulation. Simulation of the two drugs together provides molecular insight into their coexistence within the transporter and shows reasonably stable profiles, further supporting the noncompetitive nature of the two drugs. Furthermore, simulations point to a peculiar role of the vacuolar loop, which is very flexible in the binary complexes (PfCRT[Dd2]:CQ and PfCRT[Dd2]:PPQ), while having a stabilizing effect in the ternary one, altogether providing new hints on the function of the loop which has not been investigated so far. Multiple sequence alignments of different Plasmodium CRT homologs and orthologues show a high degree of conservation in the vacuolar loop, suggesting a mechanistic role [9]. The area of the transporter facing the digestive vacuole, near the vacuolar loops, was proposed as the initial binding site for CQ [42], thus supporting a possible role of the loop in the initial phase of substrate binding. Several membrane transporters are known to have extracellular loops acting as gates and taking part in the whole transport mechanisms [43], thus supporting the idea that such behavior could hold true for PfCRT.

A recent study has advocated for a possible CQ and PPQ combination therapy, which was surmised to prevent the spread of multi-drug resistant malaria given that, so far, the parasite cannot tolerate both drugs simultaneously [10]. Our study suggests that this is a too optimistic assumption and that PfCRT has the potential to evolve to overcome the CQ re-sensitizing effect, while maintaining PPQ-resistance-conferring activity. By introducing both the H97Y and the F145I mutations into PfCRT[Dd2], we generated a double mutant (PfCRT[Dd2_H97Y_F145I]) that had CQ kinetics ($K_M$: 240 ± 40 μM vs 260 ± 10 μM; $V_{max}$: 35 ± 4 pmol oocyte$^{-1}$ h$^{-1}$ vs 33 ± 1 pmol oocyte$^{-1}$ h$^{-1}$) indistinguishable from that of the parental PfCRT[Dd2]. Surprisingly, the transport activity did not follow Michaelis-Menten kinetics, but rather showed a sigmoidal response that gave a reliable fit to the Hill equation, with a Hill coefficient of 1.6 ± 0.2 (Fig 11). Similarly, a putative cooperative behavior was also observed for PPQ (Hill factor of 2.0 ± 0.3). Docking of the two drugs to the double-mutant, still producing dockings largely in agreement

with the previous results (S14 Fig), did not either support or deny a cooperative model in PfCRT$^{Dd2\_H97Y\_F145I}$, indicating that a more detailed treatment with MD simulation might be necessary to fully explore the effects of the double mutation.

Cooperativity is usually found in oligomeric protein complexes, such as hemoglobin, and occurs when binding of a substrate induces conformational changes that favor the binding of subsequent substrate molecules. There are, however, cases of cooperativity in monomeric proteins, which are explained by a single-binding site transitioning between conformations with different affinities for the substrate [44,45]. Whether such a model applies to PfCRT$^{Dd2\_H97Y\_F145I}$ remains to be seen.

Although the PPQ kinetic parameters of PfCRT$^{Dd2\_H97Y\_F145I}$ are unremarkable, with a $K_M$ of $170 \pm 20$ μM and a $V_{max}$ of $131 \pm 3$ pmol oocyte$^{-1}$ h$^{-1}$, the double mutant would outperform PfCRT$^{Dd2}$ at substrate concentrations > 200 μM. Such concentrations can be reached for CQ, and presumably for PPQ, in the digestive vacuole of *P. falciparum* parasites [19,46]. However, further experiments in the parasite are needed to investigate whether such a double mutant is viable and confers a selective advantage in the presence of CQ and PPQ.

In summary, our study demonstrates that PfCRT$^{Dd2}$ has a versatile substrate binding cavity that can accept various drugs in separate interaction domains. As a result, PfCRT$^{Dd2}$ is capable of simultaneously binding two different drugs in a ternary complex. This includes CQ and PPQ, but also CQ and quinine, and CQ and verapamil [22]. The ternary complex can be active, as shown for CQ and PPQ and for CQ and quinine, or inactive, as shown for CQ and verapamil [22]. Moreover, binding of one drug may or may not affect the affinity by which the other drug is bound. For example, CQ and PPQ interact on PfCRT$^{Dd2}$ in a noncompetitive inhibition model, whereas quinine and verapamil compete with CQ for binding to PfCRT$^{Dd2}$ in a mixed-type inhibition model [22]. In contrast, the related CQ-resistance conferring isoform PfCRT$^{7G8}$ appears to be unable to accept both CQ and PPQ simultaneously, according to molecular docking, consistent with a competitive binding model [9]. We further show that mutational changes associated with *in vivo* PPQ resistance increase the affinity and/or the $V_{max}$ for PPQ at the expanse of the CQ transport efficiency. Our finding that PfCRT can transport drugs in a cooperative manner suggests that the ability of PfCRT to adapt to changing drug pressures is much larger than initially thought and may involve variants with novel transport kinetics and broad substrate specificity.

## Materials and methods

### Ethics approval

Ethical approval of the work performed with the *Xenopus laevis* frogs was obtained from the Regierungspräsidium Karlsruhe (Aktenzeichen 35–9185.81/G-31/11 and 35-918581/G-21/23) in accordance with the German "Tierschutzgesetz".

### Reagents and radiolabeled compounds

[³H]Chloroquine (specific activity, 5–25 Ci mmol$^{-1}$) and [³H]piperaquine (specific activity, 15 Ci mmol$^{-1}$) were obtained from American Rabiolabeled Chemicals or GE Healthcare.

### Site-directed mutagenesis of *pfcrt* for expression in Xenopus laevis oocytes

A codon-optimized coding sequence of the Dd2 isoform of PfCRT (PfCRT$^{Dd2}$) was subcloned into the pSP64T expression vector [21]. To generate the necessary mutations to introduce the amino acid substitutions H97Y, F145I, M343L or G353V into the *pfcrt* sequence, the overlap extension method was used [47]. A region from the 5'-untranslated *Xenopus* β-globin sequence

to the respective 3' region, containing the full length *pfcrt* coding sequence, was amplified using the following primer pairs: for the H97Y mutation, numbers 1/4 and 2/3; for the F145I mutation, numbers 1/6 and 2/5; for the M343L mutation, numbers 1/8 and 2/7; and for the G353V mutation, numbers 1/10 and 2/9 (for primers, see S2 Table). Both PCR products were used as template for amplification of the final product using primers number 1/2. The resulting fragment was cloned into the pSP64T expression vector, and the presence of the desired mutation was confirmed by sequencing.

## Harvesting of *Xenopus* oocytes and expression of PfCRT

Adult female *X. laevis* frogs (purchased from NASCO) were anesthetized by submersion in a solution of 0.3% (w/v) 3-amino benzoate methanesulfonate for 15 min. Sections of the ovary were surgically removed and placed in $Ca^{2+}$-free ND96 buffer supplemented with 0.1% collagenase D (w/v; Roche), 0.5% BSA (w/v) and 9 mM $Na_2HPO_4$ at 18˚C for 14–16 h while gently shaking. The oocytes were then washed several times with ND96 buffer (5 mM HEPES, pH 7.5, 96 mM NaCl, 2 mM KCl, 1.8 mM $CaCl_2$, 1 mM $MgCl_2$), supplemented with 100 mg/mL gentamycin. Healthy-looking stage V-VI oocytes were manually selected and randomly grouped for injection. A codon-optimized coding sequence of PfCRT was subcloned into the pSP64T expression vector and subsequently linearized using the restriction endonuclease BamHI or SalI (New England Biolabs GmbH). RNA for injection was transcribed *in vitro* using the mMES-SAGE mMACHINE SP6 Transcription kit (Ambion), diluted with nuclease-free water to a final concentration of 0.6 μg/μL and stored at -80˚C. RNA was injected using precision-bore glass capillary tubes (3.5-inch glass capillaries; Drummond Scientific Co.), which were pulled on a vertical puller (P-87 Flaming/Brown micropipette puller, Sutter Instrument Co.) and graduated. The micropipettes were connected to a microinjector (Nanoject II Auto-Nanoliter Injector, Drummond Scientific Co.). Injection was conducted under stereomicroscopic control. 50 nL of nuclease-free water alone (control oocytes) or 50 nL of nuclease-free water containing 30 ng of RNA was injected per oocyte. Injected oocytes were kept for 72 h at 18˚C in ND96 buffer with two daily buffer changes before being used for further experiments.

## Drug transport assays

Drug transport assays were performed as previously described [22]. Briefly, measurements of radiolabeled drug uptake were made over 1 h, unless otherwise specified. Experiments were conducted at room temperature in ND96 buffer containing [³H]chloroquine (42 nM) or [³H] piperaquine (40 nM). Where specified, one or more unlabeled drugs were also present at the indicated concentrations. The direction of drug transport in the assays is from the extracellular medium (pH 6) into the oocyte cytosol (pH 7.2), which corresponds to the efflux of CQ or PPQ from the acidic digestive vacuole (pH 5.2) into the parasite cytosol (pH 7.2) [35]. The uptake assays were terminated by removing the oocytes from the reaction medium and washing them three times with 2 mL of ice-cold ND96 buffer. Each oocyte was transferred to a separate scintillation vial and lysed by the addition of 200 μL of a 5% SDS solution. The radioactivity in the sample was measured using a liquid scintillation analyser Tri-Carb 4910 TR (PerkinElmer). Water-injected oocytes were analyzed in parallel as a negative control. The buffer from each condition was measured in duplicate in parallel to each experiment to transform counts per minute into pmol of drug. Drug uptake attributable to PfCRT was determined by subtracting the background uptake from water-injected oocytes from that of PfCRT-expressing oocytes. In all cases, at least three independent experiments were performed on oocytes from different frogs, and for each condition in an experiment, measurements were made from 10 oocytes per treatment.

## The competition plot

The competition plot was performed as described elsewhere [29]. Briefly, a concentration of CQ and a concentration of PPQ were selected (using the parameters from Table 1), such that (i) the rate of PPQ transport was equivalent to the rate of CQ transport, and (ii) the transport rate for CQ (the less efficiently transported substrate) approached its $V_{max}$. The resulting concentrations for PfCRT$^{Dd2}$ were $[CQ]^{max}$ = 400 μM and $[PPQ]^{max}$ = 23.1 μM. At these concentrations of CQ and PPQ, the expected uptake is 19.8 pmol h$^{-1}$ oocyte$^{-1}$. The resulting concentrations for PfCRT$^{Dd2\_F145I}$ were $[CQ]^{max}$ = 400 μM and $[PPQ]^{max}$ = 18.2 μM. At these concentrations of CQ and PPQ, the expected uptake is 18.2 pmol h$^{-1}$ oocyte$^{-1}$. A series of reaction buffers containing CQ and PPQ at concentrations $[CQ]$ = (1—P) x $[CQ]^{max}$ and $[PPQ]$ = P x $[PPQ]^{max}$ were assembled, where P is the proportion of PPQ in the mixture. The rates of CQ and PPQ transport were measured in pairwise experiments by the addition of either [$^3$H] chloroquine or [$^3$H]piperaquine to each of the mixtures. These corresponded to values of P equal to 0, 0.25, 0.5, 0.75, 0.9 and 1. A plot of $V_{total}$ as a function of P yielded the competition plots. If both substrates bind to the same site, then a constant $V_{total}$ is expected across all P values (e.g. data lies on a horizontal line). If the substrates bind to different sites, and they inhibit each other's transport, it is expected that the curve is concave with either a minimum or a maximum.

## Equations describing models of drug competition

The following equations were adopted from Segel (1993), and the reader is referred to that text for the assumptions made in the author's approach in the derivation of each equation [31]. The concentration of substrate is denoted as [S1] and the concentration of the inhibiting substrate is denoted as [S2]. $K_{S1}$ and $K_{S2}$ are the dissociation constants for the respective substrate-protein complexes. Full mixed-type inhibition is given by Eq 1,

$$v = \frac{V_{max}\frac{[S1]}{K_{S1}}}{1 + \frac{[S1]}{K_{S1}} + \frac{[S2]}{K_{S2}} + \frac{[S1][S2]}{\alpha K_{S1} K_{S2}}}$$ (Eq1)

Partial mixed-type inhibition is given by Eq 2,

$$v = \frac{V_{max}\left(\frac{[S1]}{K_{S1}} + \frac{\beta[S1][S2]}{\alpha K_{S1} K_{S2}}\right)}{1 + \frac{[S1]}{K_{S1}} + \frac{[S2]}{K_{S2}} + \frac{[S1][S2]}{\alpha K_{S1} K_{S2}}}$$ (Eq2)

Full competitive inhibition is described by Eq 3,

$$v = \frac{V_{max}\frac{[S1]}{K_{S1}}}{1 + \frac{[S1]}{K_{S1}} + \frac{[S2]}{K_{S2}}}$$ (Eq3)

Partial competitive inhibition is described by Eq 4,

$$v = \frac{V_{max}\left(\frac{[S1]}{K_{S1}} + \frac{[S1][S2]}{\alpha K_{S1} K_{S2}}\right)}{1 + \frac{[S1]}{K_{S1}} + \frac{[S2]}{K_{S2}} + \frac{[S1][S2]}{\alpha K_{S1} K_{S2}}}$$ (Eq4)

Full noncompetitive inhibition is given by Eq 5,

$$v = \frac{V_{max}\frac{[S1]}{K_{S1}}}{1 + \frac{[S1]}{K_{S1}} + \frac{[S2]}{K_{S2}} + \frac{[S1][S2]}{K_{S1}K_{S2}}}$$ (Eq5)

Partial noncompetitive inhibition is given by Eq 6,

$$v = \frac{V_{max}\left(\frac{[S1]}{K_{S1}} + \frac{\beta[S1][S2]}{K_{S1}K_{S2}}\right)}{1 + \frac{[S1]}{K_{S1}} + \frac{[S2]}{K_{S2}} + \frac{[S1][S2]}{K_{S1}K_{S2}}}$$ (Eq6)

Full uncompetitive inhibition is described by Eq 7,

$$v = \frac{V_{max}\frac{[S1]}{K_{S1}}}{1 + \frac{[S1]}{K_{S1}} + \frac{[S1][S2]}{K_{S1}K_{S2}}}$$ (Eq7)

Partial uncompetitive inhibition is described by Eq 8,

$$v = \frac{V_{max}[S1]}{\frac{K_{S1}}{(1+\frac{\beta[S2]}{K_{S2}})} + [S1]\frac{(1+\frac{[S2]}{K_{S2}})}{(1+\frac{\beta[S2]}{K_{S2}})}}$$ (Eq8)

Ligand exclusion is given by Eq 9,

$$v = \frac{V_{max}\left(\frac{[S1]}{K_{S1}} + \frac{[S1]^2}{K_{S1}^2}\right)}{1 + \frac{2[S1]}{K_{S1}} + \frac{[S1]^2}{K_{S1}^2} + \frac{[S2]}{K_{S2}}}$$ (Eq9)

Cooperative substrate binding in which the inhibitor mimics the substrate (Cooperative Binding) is given by Eq 10,

$$v = \frac{V_{max}\left(\frac{[S1]}{K_{S1}} + \frac{[S1]^2}{aK_{S1}^2} + \frac{[S1][S2]}{aK_{S1}K_{S2}}\right)}{1 + \frac{2[S1]}{K_{S1}} + \frac{[S1]^2}{aK_{S1}^2} + \frac{2[S1][S2]}{aK_{S1}K_{S2}} + \frac{2[S2]}{K_{S2}} + \frac{[S2]^2}{aK_{S2}^2}}$$ (Eq10)

Cooperative substrate binding in which the inhibitor does not mimic the substrate (Cooperative Binding 1) is given by Eq 11,

$$v = \frac{V_{max}\left(\frac{[S1]}{K_{S1}} + \frac{[S1]^2}{aK_{S1}^2} + \frac{[S1][S2]}{K_{S1}K_{S2}}\right)}{1 + \frac{2[S1]}{K_{S1}} + \frac{[S1]^2}{aK_{S1}^2} + \frac{2[S1][S2]}{K_{S1}K_{S2}} + \frac{2[S2]}{K_{S2}} + \frac{[S2]^2}{K_{S2}^2}}$$ (Eq11)

Cooperative inhibitor binding (Cooperative Binding 2) is given by Eq 12,

$$v = \frac{V_{max}\left(\frac{[S1]}{K_{S1}} + \frac{[S1]^2}{K_{S1}^2} + \frac{[S1][S2]}{K_{S1}K_{S2}}\right)}{1 + \frac{2[S1]}{K_{S1}} + \frac{[S1]^2}{K_{S1}^2} + \frac{2[S1][S2]}{K_{S1}K_{S2}} + \frac{2[S2]}{K_{S2}} + \frac{[S2]^2}{cK_{S2}^2}}$$ (Eq12)

Two-site pure competitive inhibition is given by Eq 13,

$$v = \frac{V_{max}\left(\frac{[S1]}{K_{S1}} + \frac{[S1]^2}{aK_{S1}^2} + \frac{[S1][S2]}{bK_{S1}K_{S2}}\right)}{1 + \frac{2[S1]}{K_{S1}} + \frac{[S1]^2}{aK_{S1}^2} + \frac{2[S1][S2]}{bK_{S1}K_{S2}} + \frac{2[S2]}{K_{S2}} + \frac{[S2]^2}{cK_{S2}^2}}$$ (Eq13)

For a non-cooperative substrate in the absence of inhibitor in which the substrate reverses the effect of the inhibitor (Substrate non-cooperative), Eq 14 is given,

$$v = \frac{V_{max}\left(\frac{[S1]}{K_{S1}} + \frac{[S1]^2}{K_{S1}^2} + \frac{[S1][S2]}{\alpha K_{S1} K_{S2}} + \frac{[S1]^2[S2]}{\alpha K_{S1}^2 K_{S2}}\right)}{1 + \frac{2[S1]}{K_{S1}} + \frac{[S1]^2}{K_{S1}^2} + \frac{2[S1][S2]}{\alpha K_{S1} K_{S2}} + \frac{[S2]}{K_{S2}} + \frac{[S1]^2[S2]}{\alpha K_{S1}^2 K_{S2}}} \tag{Eq14}$$

For a cooperative substrate in the absence of inhibitor in which the substrate reverses the effect of the inhibitor (Substrate cooperative), Eq 15 is given,

$$v = \frac{V_{max}\left(\frac{[S1]}{K_{S1}} + \frac{[S1]^2}{a K_{S1}^2} + \frac{[S1][S2]}{\alpha K_{S1} K_{S2}} + \frac{[S1]^2[S2]}{a\alpha K_{S1}^2 K_{S2}}\right)}{1 + \frac{2[S1]}{K_{S1}} + \frac{[S1]^2}{a K_{S1}^2} + \frac{2[S1][S2]}{\alpha K_{S1} K_{S2}} + \frac{[S2]}{K_{S2}} + \frac{[S1]^2[S2]}{a\alpha K_{S1}^2 K_{S2}}} \tag{Eq15}$$

A system in which the inhibitor eliminates substrate cooperativity is given by Eq 16,

$$v = \frac{V_{max}\left(\frac{[S1]}{K_{S1}} + \frac{[S1]^2}{a K_{S1}^2} + \frac{[S1][S2]}{K_{S1} K_{S2}} + \frac{[S1]^2[S2]}{K_{S1}^2 K_{S2}}\right)}{1 + \frac{2[S1]}{K_{S1}} + \frac{[S1]^2}{a K_{S1}^2} + \frac{2[S1][S2]}{K_{S1} K_{S2}} + \frac{[S2]}{K_{S2}} + \frac{[S1]^2[S2]}{K_{S1}^2 K_{S2}}} \tag{Eq16}$$

## Discrimination between drug competition models and statistical analyses of the kinetic data

Analyses of the kinetic data were performed using SigmaPlot version 13.0 and Python. The drug competition models were globally fit to the kinetic data using the least-squares method and ranked according to their corrected Akaike information criterion difference ($\Delta AIC_C$) and their Akaike weight [30], using Python. The two top-ranked models were compared using an F-test.

The corrected Akaike information criterion ($AIC_C$) of each model was calculated according to Eq 17,

$$AIC_C = n \times ln\left(\frac{RSS}{n}\right) + 2 \times K + \frac{2 \times K \times (K+1)}{n - K - 1} \tag{Eq17}$$

where RSS is the residual sum of squares,

$$RSS = \sum_{i=1}^{n} (y_i - \hat{y}_i)^2 \tag{Eq18}$$

n is the total number of measurements used to perform the global fit, $y_i$ are the experimentally measured values, $\hat{y}_i$ are the values predicted by the model, and K is the number of parameters in the model. The $\Delta AIC_C$ of the ith model ($\Delta_i$) was calculated according to Eq 19,

$$\Delta_i = AIC_C^i - AIC_C^{min} \tag{Eq19}$$

where the $AIC_C^{min}$ is the smallest $AIC_C$ value of all of the models tested.

For a more detailed evaluation of the plausibility of the models, the Akaike weight ($w_i$) was calculated according to Eq 20,

$$w_i = \frac{e^{-\frac{1}{2} \times \Delta_i}}{\sum_{j=1}^{J} e^{-\frac{1}{2} \times \Delta_j}} \tag{Eq20}$$

where $\Delta_i$ is as described above, $\Delta_j$ is the $AIC_C$ difference of the jth model tested, and J is the total number of models investigated. $w_i$ can range from 0 to 1 and reports the plausibility of

the of the ith model. The model with the highest Akaike weight is more likely to be correct, and the ratio of the Akaike weight of two models reports how much more likely one model is with respect to the other.

The F statistic for the selection between the two best ranking models was calculated according to Eq 21, and the corresponding p value was obtained using R.

$$F = \frac{RSS_1 - RSS_2}{RSS_2} \times \frac{df_2}{df_1 - df_2}$$

(Eq21)

Here, $RSS_1$ is the residual sum of squares of the simpler model (the one with less parameters), and $RSS_2$ is the residual sum of squares of the more complex model (the one with more parameters). $df_1$ and $df_2$ are the degrees of freedom of the simpler and more complex model, respectively.

## Determination of the value of the parameter α

As can be seen in Eqs 2 and 6, the difference between Mixed-type (Partial) and Noncompetitive (Partial) inhibition models is the value of α. This factor reflects how much the affinity of the transporter for one drug changes when the other drug is already bound in the protein cavity. In the former model, $\alpha > 0$ and $\alpha \neq 1$. In the latter model, it could be thought of as if $\alpha = 1$. Because of this, discriminating between the two models also means determining the value of α. In Mixed-type (Partial) inhibition, the apparent maximal velocity ($V_{max,app}$) is given by Eq 22,

$$V_{max,app} = V_{max} \frac{(1 + \frac{\beta[S_2]}{\alpha K_{S2}})}{(1 + \frac{[S_2]}{\alpha K_{S2}})}$$

(Eq22)

Multiplying nominator and denominator by $\alpha K_{S2}$, Eq 23 is obtained,

$$V_{max,app} = V_{max} \frac{(\alpha K_{S2} + \beta[S_2])}{(\alpha K_{S2} + [S_2])}$$

(Eq23)

This equation has the form of a hyperbole when plotting $V_{max,app}$ as a function of inhibitor concentration ([S2]). $V_{max,app}$ is the maximal rate of substrate transport at a certain inhibitor concentration. The half-maximal inhibitory concentration is given by the term in the denominator different from [S2], which in this case is $\alpha K_{S2}$. This means that by plotting $V_{max,app}$ as a function of [S2], the value of $\alpha K_{S2}$ can be derived after fitting the curve to Eq 23.

Additionally, the apparent Michaelis constant ($K_{M,app}$) is given by Eq 24,

$$K_{M,app} = K_M \frac{(1 + \frac{[S_2]}{K_{S2}})}{(1 + \frac{[S_2]}{\alpha K_{S2}})}$$

(Eq24)

Dividing Eq 22 by Eq 24,

$$\frac{V_{max,app}}{K_{M,app}} = \frac{V_{max}}{K_M} \frac{(1 + \frac{\beta[S_2]}{\alpha K_{S2}})}{(1 + \frac{[S_2]}{K_{S2}})}$$

Multiplying numerator and denominator by $K_{S2}$, Eq 25 is obtained,

$$\frac{V_{max,app}}{K_{M,app}} = \frac{V_{max}}{K_M} \frac{(K_{S2} + \frac{\beta[S_2]}{\alpha})}{(K_{S2} + [S_2])}$$

(Eq25)

The half-maximal inhibitory concentration derived from this hyperbolic equation is equal to $K_{S2}$. By plotting $V_{max,app}/K_{M,app}$ as a function of [S2], one can derive $K_{S2}$ after fitting the curve to Eq 25. The value of $\alpha$ can then be calculated dividing $\alpha K_{S2}$ by $K_{S2}$.

## Western analysis of oocyte lysates

Western analyses were performed as described [48]. Briefly, oocyte lysates were prepared 72 h after RNA injection (the same day when drug transport assays were performed), by addition of 20 μL of homogenization buffer (10 mM HEPES, pH 7.4, 150 mM NaCl, 1% NP-40, 0.5% sodium deoxycholate, 0.1% SDS) supplemented with a protease inhibitor cocktail (Complete, Roche Applied Science) per oocyte. Cellular debris was removed by centrifugation at 16000 x *g* for 15 min at 4˚C. Supernatants were mixed with 2x sample buffer (250 mM Tris, pH 6.8, 3% SDS, 20% glycerol, 0.1% bromophenol blue), boiled at 95˚C for 3 min and stored at -20˚C. Lysates were then size-fractionated using 10% SDS-PAGE and transferred to a polyvinylidene difluoride membrane. After transfer, the membrane was cut in two halves below the 55 kD marker around 50 kD (see S1 Fig). The upper half was incubated with a monoclonal mouse anti-α-tubulin antiserum (1:1000 dilution; clone B-5-1-2) and a goat anti-mouse POD anti-body (1:10000 dilution; Jackson Immunoresearch Laboratories). The lower half was incubated with a guinea pig anti-PfCRT antiserum (raised against the N terminus of PfCRT (MKFASKKNNQKNSSK); 1:1000 dilution; Eurogentec) and a donkey anti-guinea pig POD antibody (1:10000 dilution; Jackson Immunoresearch Laboratories). All antibodies were diluted in 1% (w/v) BSA in PBS. After overnight incubation, the two halves were placed on a digital blot scanner (C-DiGit) and the the signal intensities were captured. The signal intensities were quantified using Image Studio Digits version 4.0 (LiCor).

## Immunofluorescence assay of oocytes

Immunofluorescence assays were performed as previously described [48]. Briefly, three days after RNA injection, oocytes were fixed by incubation with 4% (v/v) paraformaldehyde in PBS for 4 h at room temperature. Fixed cells were washed three times with 3% (w/v) BSA in PBS. After washing three times with 3% (w/v) BSA in PBS, oocytes were incubated with guinea pig anti-PfCRT antiserum (1:500 dilution) overnight at 4˚C. After washing again for three times, the secondary antibody anti-rabbit Alexa Fluor 488 (dilution 1:1000) was added and allowed to incubate for 45 min. After 3 subsequent washing steps, the antibody serving as an internal control, wheat germ agglutinin Alexa Fluor 633 (5 μg mL$^{-1}$), was added and the oocytes were incubated for 10 min at 4˚C. After three washing steps, the oocytes were analysed by fluorescence microscopy. Images were taken with a Zeiss LSM 510 confocal microscope and processed with the Fiji program.

## Molecular docking studies

The cryo-EM structure of the PfCRT 7G8 isoform (PDB ID: 6UKJ; 3.2 Å) [9] was used as a template for homology modelling. The SWISS-MODEL Protein Modelling server [34] (https://swissmodel.expasy.org/) was used to generate the model. The structure of the PfCRT$^{Dd2}$ model was then prepared using the Protein Preparation Wizard in Maestro (Schrödinger Release 2021–4) [49] and protonated at pH 6.0 using PROPKA [50]. Prior to docking calculations, the two drugs, CQ and PPQ, were prepared with LigPrep and ionized at pH 6.0 using Epik [51]. Induced-fit docking (IFD) [52] was performed using the default options if not specified otherwise. The same docking box was used for both compounds and built by selecting Q235, S90, Q156, L221, W352, L83, and F145 as centroids. For both CQ and PPQ, 10 poses were generated. The first docking pose of PPQ and the third pose of CQ were then merged

and embedded into a POPC (1-palmitoyl-2-oleoyl-phosphatidylcholine) lipid bilayer, built with the CHARMM GUI webserver [53] using the Amber Lipid14 force field [54]. The systems were then solvated in a periodic box of TIP3P [55] water molecules and neutralized at an ion concentration of 150 mM NaCl. The GAFF [56] force field was used for parameterizing the two compounds along with AM1-BCC for assigning partial charges [57]. The Amber ff14SB force field [56] was used to assign protein parameters. MD simulations were run using the Amber20 software [58]. The systems were first energy minimized with 10 consecutive runs (1000 steps each) of decreasing restraints from 1000 to 0.01 kcal mol$^{-1}$ Å$^{-2}$ applied to the heavy atoms and an additional 1000 steps without restraints. Heating was performed in two stages of 200 ps: first to 100 K using restraints of 100 kcal mol$^{-1}$ Å$^{-2}$ with the Langevin thermostat (NVT), and then up to 310 K with restraints of 5 kcal mol$^{-1}$ Å$^{-2}$. A 4 ns equilibration with restraints of 5 kcal mol$^{-1}$ Å$^{-2}$ was then performed. Additionally, 10 consecutive simulations (5 ns each) without restraints were performed to equilibrate the system's periodic boundary dimensions. Finally, a production of 1 μs under the NPT ensemble (Langevin thermostat at 310 K with a Berendsen barostat at 1 bar) was run. A time step of 2 fs was used for all simulations and bonds with hydrogens were constrained using the SHAKE algorithm.

## Statistics and reproducibility

Statistical analyses were performed using Sigma Plot (v.14.5, Systat) software. Statistical significance was determined using the paired Student's *t* test, or the F-test where appropriate. p values < 0.05 were considered significant. The number of independent biological replicates is indicated in the main text and/or the figure legends. If independent data points were averaged, then the mean ± SEM is shown. Independent biological replicates are defined as experiments using oocytes from a different frog. Per condition and independent biological replicate, at least 10 oocytes were investigated. The CQ transport is shown throughout this manuscript as the mean ± SEM of 3 to 8 independent biological replicates, within which measurements were made from 10 oocytes per condition. The original data underpinning this study can be found in the source data file. The experimental data were uploaded to Dryad https://doi.org/10.5061/dryad.prr4xgxr5 and Zenodo https://doi.org/10.5281/zenodo.7900741 online repositories [59,60].

## Dryad DOI

https://doi.org/10.5061/dryad.prr4xgxr5 [59]; https://doi.org/10.5281/zenodo.7900741 [60]

## Supporting information

**S1 Table. Glide SP docking scores (kcal/mol) of the 10 poses generated with the IFD protocol for the binding of CQ or PPQ in the cavity of PfCRT$^{Dd2}$, PfCRT$^{Dd2\_F145I}$, PfCRT$^{Dd2\_H97Y}$, PfCRT$^{Dd2\_M343L}$, PfCRT$^{Dd2\_G353V}$, PfCRT$^{Dd2\_H97Y\_F145I}$ and PfCRT$^{7G8}$.**
(PDF)

**S2 Table. Primers used for the mutagenesis of *pfcrt* onto the codon-optimized *pfcrt*$^{Dd2}$ sequence in the pSP64T vector backbone.**
(PDF)

**S3 Table. Original data underpinning this study.**
(XLSX)

**S1 Fig. Localization of PfCRT variants on the oocyte plasma membrane.** Immunofluorescence images of fixed PfCRT -expressing oocytes and water-injected control oocytes. *First*

*panel from the left*, fluorescence image of PfCRT using a specific guinea pig antiserum primary antibody (α-PfCRT) and the Alexa Fluor 633 anti-guinea pig secondary antibody. *Second panel*, fluorescence image of wheat germ agglutinin (WGA) conjugated to Alexa Fluor 488. *Third panel*, differential interference contrast (DIC) image. *Fourth panel*, overlay. Scale bar, 135 μm. For visualization purposes, contrast was enhanced.
(TIF)

**S2 Fig. PfCRT variants are expressed at similar leves in the oocyte plasma membrane.** Western blot analyses of total lysates from oocytes expressing PfCRT variants and water-injected oocytes, using the polyclonal guinea pig antiserum specific to PfCRT and, as a loading control, mouse monoclonal anti-α-tubulin antibody. The luminescence signals from independent Western blot analyses were quantified, yielding the PfCRT expression levels relative to the internal standard α-tubulin. A box plot analysis is overlaid over the individual data points (independent biological replicates) with the median (black line), mean (red line), and 25 and 75% quartile ranges being shown. Statistical significance was assessed using the Brown-Forsythe ANOVA and found to be p = 0.795.
(TIF)

**S3 Fig. Verapamil inhibits PfCRT-mediated transport of CQ and PPQ.** The inhibitory effect of adding 100 μM verapamil (VP) to an extracellular medium containing a total concentration of 50 μM *A*, chloroquine (CQ) or *B,* piperaquine (PPQ), was evaluated in water-injected (control) and oocytes expressing PfCRT$^{Dd2}$ (Dd2), PfCRT$^{Dd2\_H97Y}$ (H97Y), PfCRT$^{Dd2\_F145I}$ (F145I), PfCRT$^{Dd2\_M343L}$ (M343L) or PfCRT$^{Dd2\_G353V}$ (G353V). *Left panels*, data is shown as individual biological replicates (coloured symbols) overlaid onto bars representing the Mean ± S.E.M. (error bars). A one-tailed Student's t-test was performed between untreated (no VP) and treated (+ 100 μM VP) samples, and the obtained *p* values are shown. *Right panels*, the uptake in VP-treated samples was divided by the uptake in their corresponding untreated pairs and subtracted from 100% to get the percentage of uptake inhibition mediated by the presence of 100 μM verapamil in the extracellular medium. Error bars are S.E.
(TIF)

**S4 Fig. Ramachandran plot of the PfCRT$^{Dd2}$ model.** The structure validation was performed with MolProbity, implemented in the SWISS-MODEL server [34]. 96.64% of the residues are positioned in the favored regions and only 0.28% are in outlier regions. The residues, indicated by dots, are colored according to the QMEAN parameter, indicating the residue quality. The color scale goes from red (bad quality) to blue (good quality).
(TIF)

**S5 Fig. Docking results of CQ and PPQ to PfCRT$^{Dd2}$.** A blue-to-red color scale shows the generated docking poses for CQ (A-C) and PPQ (D-F), ranked from the best (blue) to the worst (red) docking score.
(TIF)

**S6 Fig. Replacement of specific amino acid residues with Ala hampers CQ transport.** The PfCRT-mediated transport of CQ was measured in oocytes expressing PfCRT$^{Dd2}$, PfCRT$^{Dd2\_V141A}$, PfCRT$^{Dd2\_S257A}$ or PfCRT$^{Dd2\_I260A}$. A box plot analysis is overlaid over the individual data points (independent biological replicates) with the median (black line), mean (red line), and 25 and 75% quartile ranges being shown. Multiple comparisons versus the PfCRT$^{Dd2}$ control group was performed (Holm-Sidak ANOVA).
(TIF)

**S7 Fig. Root-mean-square deviation (RMSD) analysis of CQ and PPQ in complex with PfCRT<sup>Dd2</sup>.** A, B. CQ (A) and PPQ (B) RMSD in their respective complexes with PfCRT<sup>Dd2</sup> (PfCRT<sup>Dd2</sup>:CQ and PfCRT<sup>Dd2</sup>:PPQ). C, D. CQ (C) and PPQ (D) RMSD in the PfCRT<sup>Dd2</sup>:CQ: PPQ complex.
(TIF)

**S8 Fig. MD simulations of the PfCRT<sup>Dd2</sup>:CQ and PfCRT<sup>Dd2</sup>:PPQ complexes.** The last frame from the MD simulations is shown and superimposed on the original docking pose for (A) CQ, and (B) PPQ. The docking and the molecular dynamics final poses are shown in green and yellow, respectively, for CQ, and cyan and blue, respectively, for PPQ. The protein and the amino acid residues that interact with ligands are shown in light blue.
(TIF)

**S9 Fig. Loop arrangements during the MD simulations.** A blue-to-red color scale is used to indicate the movement of the loop along the simulation trajectories, as highlighted by the arrows on the schematic figures. Loop rearrangements are shown for the (A) PfCRT<sup>Dd2</sup>:CQ, (B) PfCRT<sup>Dd2</sup>:PPQ, and (C) PfCRT<sup>Dd2</sup>:CQ:PPQ complexes. Starting and final binding poses (encircled) are shown for CQ in panel (A).
(TIF)

**S10 Fig. Most representative docking results of CQ and PPQ to PfCRT<sup>Dd2_F145I</sup>.** A blue-to-red color scale shows the generated docking poses for CQ (**A-B**) and PPQ (**C-F**), ranked from the best (blue) to the worst (red) docking score. The F145I mutation is indicated as a sphere in cyan.
(TIF)

**S11 Fig. Most representative docking results of CQ and PPQ to PfCRT<sup>Dd2_H97Y</sup>.** A blue-to-red color scale shows the generated docking poses for CQ (**A-B**) and PPQ (**C-E**), ranked from the best (blue) to the worst (red) docking score. The H97Y mutation is indicated as a sphere in cyan.
(TIF)

**S12 Fig. Most representative docking results of CQ and PPQ to PfCRT<sup>Dd2_G353V</sup>.** A blue-to-red color scale shows the generated docking poses for CQ (**A-C**) and PPQ (**D-F**), ranked from the best (blue) to the worst (red) docking score. The G353V mutation is shown as a sphere in cyan.
(TIF)

**S13 Fig. Most representative docking results of CQ and PPQ to PfCRT<sup>Dd2_M343L</sup>.** A blue-to-red color scale shows the generated docking poses for CQ (**A-C**) and PPQ (D-E), ranked from the best (blue) to the worst (red) docking score. The M343L mutation is shown as a sphere in cyan.
(TIF)

**S14 Fig. Most representative docking results of CQ and PPQ to PfCRT<sup>Dd2_H97Y_F145I</sup>.** A blue-to-red color scale shows the generated docking poses for CQ (**A-B**) and PPQ (**C-D**), ranked from the best (blue) to the worst (red) docking score. The H97Y and F145I mutations are shown as a sphere in cyan.
(TIF)

## Acknowledgments

We thank Marina Müller and Atdhe Kernaja for excellent technical assistance. We thank Augusto Masetti for his invaluable help with the Python code.

## Author Contributions

**Conceptualization:** Guillermo M. Gomez, Cecilia P. Sanchez, Rebecca C. Wade, Michael Lanzer.

**Formal analysis:** Guillermo M. Gomez, Giulia D'Arrigo, Cecilia P. Sanchez.

**Funding acquisition:** Rebecca C. Wade, Michael Lanzer.

**Investigation:** Guillermo M. Gomez, Giulia D'Arrigo, Cecilia P. Sanchez, Fiona Berger.

**Methodology:** Guillermo M. Gomez, Giulia D'Arrigo.

**Resources:** Cecilia P. Sanchez, Rebecca C. Wade, Michael Lanzer.

**Supervision:** Rebecca C. Wade, Michael Lanzer.

**Visualization:** Guillermo M. Gomez, Giulia D'Arrigo.

**Writing – original draft:** Guillermo M. Gomez, Giulia D'Arrigo, Michael Lanzer.

**Writing – review & editing:** Guillermo M. Gomez, Giulia D'Arrigo, Cecilia P. Sanchez, Fiona Berger, Rebecca C. Wade, Michael Lanzer.

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
