## [Decision Letter · Decision Letter 0]

12 Mar 2023

Dear Prof. Lanzer,

Thank you very much for submitting your manuscript "PfCRT mutations conferring piperaquine resistance in falciparum malaria shape the kinetics of quinoline drug binding and transport" for consideration at PLOS Pathogens. As with all papers reviewed by the journal, your manuscript was reviewed by members of the editorial board and by several independent reviewers. In light of the reviews (below this email), we would like to invite the resubmission of a significantly-revised version that takes into account the reviewers' comments.

This manuscript has now been evaluated by three researchers with expertise in different aspects of this study and the broad field of antimalarial drug resistance. All three reviewers provide favorable comments and highlight the interest of this study and the high quality of the manuscript. Some concerns are raised, notably reviewer 2 who requests additional data, notably the inclusion of more controls. Reviewer 2 also was unable to access some data in a repository. The authors are encouraged to resubmit this manuscript once the reviewer comments have been addressed, which is likely to include additional experimentation. 

We cannot make any decision about publication until we have seen the revised manuscript and your response to the reviewers' comments. Your revised manuscript is also likely to be sent to reviewers for further evaluation.

Sincerely,

David A. Fidock

Guest Editor

PLOS Pathogens

Kami Kim

Section Editor

PLOS Pathogens

Kasturi Haldar

Editor-in-Chief

PLOS Pathogens

orcid.org/0000-0001-5065-158X

Michael Malim

Editor-in-Chief

PLOS Pathogens

orcid.org/0000-0002-7699-2064

Reviewer's Responses to Questions

**Part I - Summary**

Reviewer #1: This study uses the competition kinetics of chloroquine (CQ) and piperaquine (PPQ0, together with docking and molecular dynamics simulations, to investigate the mechanism of various mutations in the PfCRT conferring resistances to both drugs. CQ and PPQ competition assays suggest a noncompetitive inhibition mechanism. The results show that PfCRT can accept simultaneously both CQ and PPQ at distinct but allosterically interacting sites. Structural simulation analyses also appeared to support the noncompetitive nature of the two drugs. Additionally, PfCRT from CQ-resistant parasites is better at transporting CQ than are PfCRT variants from PPQ-resistant parasites, and vice versa. These observations provide mechanistic insight into the PfCRT structure and transport functions for the two drugs. Drug resistance in malaria and PfCRT-mediated resistance has been and will continue to be ‘hot topics’ in malaria research. This study will attract the attention of many scientists in the research fields of malaria and other parasites.

One negative part of this study is that the analyses were all performed in X. laevis expression system, not the parasite itself. Studies have shown that some PPQ-resistant mutations in Pf parasites are dependent on the parasite’s genetic background. Maybe the authors can address this point in the Discussion.

Reviewer #2: Mutations in PfCRT play a crucial role in altered drug susceptibility of the malaria parasite to historical antimalarials such as chloroquine (CQ) and more recently to current front-line components of antimalarial treatments such as piperaquine (PPQ). Understanding how PfCRT mutations confer resistance to PPQ may be critical for combatting emerging resistance. Previous studies using the xenopus laevis oocyte system by these authors and others have provided convincing evidence that mutant forms of PfCRT transport CQ with varying kinetics (e.g. PMID: 24728833), and that CQ-resistant PfCRT is able to transport a range of different substrates, likely via interactions at multiple binding sites (PMID: 25378409). Here the authors leverage the same approach and analytical techniques to investigate PfCRT interactions with Piperaquine and chloroquine.

In contrast to prior literature, this study indicates that PPQ is a substrate of pre-existing PfCRT variants (PfCRT-Dd2) which does not itself confer resistance to PPQ. Furthermore the study indicates that mutations that confer PPQ resistance and a corresponding loss of CQ resistance, retain appreciable levels of CQ transport activity. These findings do not correspond precisely with previous observations in parasites, where correlations between PPQ accumulation and resistance phenotypes are weak. However the authors make a reasonable argument that nuanced differences in kinetics of drug transport do indeed account for the altered PPQ and CQ susceptibility conferred by these mutations. While the data supporting these claims appear reasonable, additional controls are required to validate the findings, and a more transparent approach to correlating transport with resistance phenotype is required. The analysis also lacks an account of how these mutations influence the physiological role of the transporter, which may itself be an important contribution to the resistance phenotype. It is also puzzling that the transport rate of PPQ reported here is vastly higher than for CQ, and the authors should be sure that all appropriate controls have been included to ensure that only the specific PfCRT-mediated PPQ transport is being detected. The authors also examine the transport properties of an engineered double-mutant (Dd2-H97V-F145I) with unusual kinetic properties, which may or may not confer cross-resistance to both CQ and PPQ. Additional experiments are required to fully substantiate the claims and physiological relevance of this finding.

Building on previous kinetic studies of CQ and quinine interactions, the authors next examine the inhibition of CQ transport by PPQ, and PPQ transport by CQ in PfCRT-Dd2 expressing oocytes in order to examine the mechanism of PPQ and CQ interactions with PfCRT. This aspect of the work is applies appropriate analyses and experiments to establish that PPQ and CQ interact with PfCRT at distinct sites within the substrate binding pocket and influence transport of one another largely by noncompetitive mechanisms. The same mechanisms of interaction were observed in a PPQ resistant form of PfCRT. With the contingent uncertainty around reported PPQ transport rates aside, this aspect of the study is complete and comprehensive. Molecular dynamics simulations and docking studies provide corroborating evidence suggesting plausible sites at which both CQ and PPQ could interact with PfCRT-Dd2. Overall this aspect of the study builds on previous observations by the same group and contribute to the understanding of PfCRT as accommodating a range of substrates via multiple interaction sites.

Overall the manuscript is well written, the overall rational for the study and analysis are sound and treats prior work fairly and appropriately, however additional data would further support the claims made.

Reviewer #3: PfCRT harbors mutations that mediate the efflux of chloroquine (CQ) from the digestive vacuole where the drug action mechanism occurs. Additional mutations on the Southeast Asian CQ-resistant PfCRT Dd2 variant have recently been evolved in response to drug pressure which in turn mediating piperaquine (PPQ) resistance while re-sensitizing CQ. In this manuscript, Gomez et al. have performed transport kinetic studies, competition plot experiments and molecular dynamics simulations to describe how additional mutations associated with PPQ resistance alter transport kinetics of CQ and PPQ, and both drugs can allosterically bind at distinct sites within the drug-binding cavity of PfCRT. Moreover, CQ and PPQ appear to act as partial noncompetitive inhibitors as opposed to competitive inhibitors as previously suggested. The authors have further engineered PfCRT by introducing two mutations associated with PPQ resistance which enhanced transport efficiency for both drugs, suggesting that the rationale of using both CQ and PPQ as a combination therapy may not be the most ideal for malaria treatment.

Understanding how CQ and PPQ bind to PfCRT, and how mutations in PfCRT influences drug transport are significantly important. The findings from this manuscript would provide insight into the molecular mechanism utilized by PfCRT to mediate CQ and PPQ resistance. The manuscript is well-written, and the authors have provided thorough explanations on transport kinetics and competition plot experiments.

I have the following questions and comments:

In the competition plot experiment (Figure 3), I would be inclined to see the competition plot for the interaction of PfCRT Dd2 with CQ/PPQ and Verapamil, a mixed-type inhibitor. In my opinion, this would provide a better understanding on the differences between noncompetitive and mixed-type inhibition models.

The authors should comment on their rationale for the use of POPC as a choice of lipid for the simulations.

The study reports an intriguing observation on a loop rearrangement in the digestive vacuole upon drug bindings over the simulation time (Fig. 9B and S7 Fig). I’ve always wondered if the loop in question is involved in the gating of substrate/drug through an alternating-access mechanism or if it has other regulatory roles. In my opinion, this phenomenon should be further discussed. Is there perhaps any known transporter with a similar structural feature anchoring substrate and stabilizing the complexes?

The term ‘extracellular loop’ may be confusing, as it is on the digestive vacuole side. I suggest the authors use a different term or specify the location of loop between TM 7 and 8.

The docking was performed on a homology model. Have the authors tried docking/simulation for CQ and PPQ on the cryo-EM structure of the PfCRT 7G8 isoform? I think a comparison between the two isoforms could further support the locations where CQ and PPQ bind in the PfCRT CQ-resistant isoforms and enhance robustness of the Dd2 homology models.

**Part II – Major Issues: Key Experiments Required for Acceptance**

Reviewer #1: (No Response)

Reviewer #2: Figure 1 shows that PPQ transport rates were consistently higher than for CQ. Given that PfCRT-Dd2 bearing parasites have no difference in PPQ accumulation compared to CQ sensitive PfCRT-3D7 parasites (e.g. PMID: 36306287), how do the authors explain this apparent high rate of PPQ transport? Were the figures in panel A and B conducted within oocytes from the same batch? The authors should include a wild-type PfCRT control (or another unrelated transporter not involved in drug transport) in Figure 1 that would not be expected to transport CQ or PPQ for additional validation. The relatively small difference in drug uptake between water-injected and PfCRT-expressing oocytes compared to previous studies (e.g. PMID:25378409) is concerning, and suggests a low signal to noise level that would reduce the sensitivity of subsequent kinetics. The findings of high levels of PPQ transport via PfCRT-Dd2 also contradict recent observations indicating that Dd2 PfCRT is unable to transport PPQ (PMID: 35130315). Given these discrepancies the authors should seek corroborating evidence that the rates of PPQ transported reported here are due entirely to specific transport of the drug by PfCRT-Dd2.

The authors assert that relatively subtle differences in drug transport kinetics explain differing levels of CQ and PPQ resistance in parasites bearing existing and novel PfCRT mutations. This conclusion relies largely on a Principal Component Analysis of transport data for five PfCRT variants along with corresponding drug-resistance data from other sources. Critical controls have not been conducted (i.e. expression levels for all variants tested by western blot and inclusion of a bona fide CQ and PPQ-sensitive PfCRT variant), and the PSA plot is unclear (see part III below). The findings also contradict recent observations indicating that transport alone cannot account for the resistance phenotype of these mutations in isogenic parasites (PMID: 36306287) and no discussion of the impact of these mutations on the physiological role of PfCRT is attempted.

The transport properties of a double-mutant (PfCRT-Dd2-H97Y-F145I) could render both CQ and PPQ resistance, however at many concentrations this variant has lower drug transport rates than other variants tested. Were CQ and PPQ kinetics experiments conducted in parallel with PfCRT Dd2-expressing oocytes, or is the dotted line in figure 10 data from other experiments? Ideally these comparisons would be made in pairwise experiments with oocytes from the same batch. The expression levels of this variant were also not assessed. Both the H97Y and F145I mutations incur high fitness costs in multiple PfCRT and genome backgrounds of the parasite (PMID: 30115924, 36082431 36306287, 35130315), suggesting that a double mutant would be exceedingly unlikely to be viable. If the authors wish to make this a key finding of the study, efforts should be made to test the contribution of this variant to altered drug susceptibility in parasites (e.g. via episomal expression as previously conducted by this group PMID: 24728833).

Reviewer #3: The molecular dynamics simulations and docking studies have demonstrated that CQ likely binds to a region that includes TM 1, 2, 3 and 7 via a salt bridge with E75, a hydrogen bond with S257 and hydrophobic contacts with I260, V141 and F145. Given the rather poor docking score for CQ, I think that characterizing both binding and transport properties of these residues following mutagenesis could increase the confidence of the molecular docking results.

**Part III – Minor Issues: Editorial and Data Presentation Modifications**

Reviewer #1: A PfCRTDd2_H97Y_F145I double mutant was also generated and tested for responses to CQ and PPQ. The curves showed a sigmoidal response, reflecting the different impacts of the two mutations on the drugs. Is it possible that at lower drug concentrations, the H97Y contributed more to the changed transport activity, whereas at higher concentrations, the F145I was the major player?

Isoborogram is a widely used method for evaluating drug interactions. Did the authors try this method?

Line 132, ‘significant’ more CQ’. But there is no significant test.

Figures 2A and 2B are not cited in the text.

Figure 2C legend, please explain the arrows and the why the specific parameters used in the PCA. Why and how IC50 for CQ, and PSA and CAR for PPQ were chosen for the analysis?

Fig. S4. Can the different poses be displayed in sequential figures? It is difficult to see the changes in the mixed poses in the same figure.

Fig. S5, spell out RMSD.

Reviewer #2: Table 1 and subsequent tables should include the number of replicates for each construct/substrate in addition to mean ± SEM.

The PSA plot and analysis in Figure 2 is not a sufficiently clear illustration of the relationships between resistance and drug transport activity. No grouping of PPQ resistant variants is evident, and it is not clear whether the vectors displayed are scaled by weighting or significance. PSAs on small datasets can also be misleading by obscuring outlier values that may drive overall trends and can overstate the contribution of variables with small ranges. If the authors wish to establish a correlation between measures of in vitro drug susceptibility and drug transport properties such as Vmax or Km, (or a composite metric such as Vmax/Km) for various PfCRT variants, it would be preferable to present these correlations directly. The cellular resistance data for H97Y is taken from a strain with a different genetic background (PH1263-C) to the other variants (Dd2) and may confound the analysis. More up to date data including PPQ and CQ cellular accumulation ratios published in Okombo & Mok et al 2022 should also be included (PMID: 36306287).

Supplementary figure S1 panel A could include a merge of WGA, PfCRT and DIC to illustrate that PfCRT expression is indeed at the surface of the oocytes.

Supplementary figure S1B indicates large discrepancies in expression levels in oocytes from different batches, and some experiments resulted in markedly higher expression levels of one genotype compared to another. What efforts were made to control for expression levels in drug transport experiments comparing different PfCRT variants? Were transport rates normalized to internal controls? How were differing expression levels accounted for between biological replicates? If any normalizations were performed, the authors should clearly state so in the methods and figure legends.

It is unclear why certain docking representations were included in Figure 6 – was this due to proximity to ‘interesting’ residues, or due to calculated affinity? Are the same binding modes for PPQ and CQ present in PPQ resistant PfCRT variants? How do the mutations impact docked affinity? How do the identified CQ and PPQ binding sites compare with other recent PfCRT modelling studies (i.e. PMID: 36800498)? Molecular docking with the H97H-F145I double-mutant could also shed light on the possibility of cooperativity in this transporter.

Reviewer #3: The legends for Figure 8-10 need to be correctly labeled.

Figure 2A, B needs to be cited in the text.

The full name for ‘PC’ on the axis in Figure 2C should be indicated in the figure legend.

In all Figure legends, authors used abbreviations for CQ and PPQ but not in Figure 9. I advocate for consistency.

In Figure 9, the molecular dynamics poses for CQ and PPQ should be colored differently.

PLOS authors have the option to publish the peer review history of their article (what does this mean?). If published, this will include your full peer review and any attached files.

Reviewer #1: No

Reviewer #2: No

Reviewer #3: No
---

## [Editor Report · Decision Letter 1]

21 May 2023

Dear Prof. Lanzer,

We are pleased to inform you that your manuscript 'PfCRT mutations conferring piperaquine resistance in falciparum malaria shape the kinetics of quinoline drug binding and transport' has been provisionally accepted for publication in PLOS Pathogens.

We appreciate the care with which you addressed the first round of reviews. Your revised manuscript has been reviewed by Reviewer 2, who is fully satisfied with the changes made and who has no further requests. A member of the Editorial team has also reviewed your revision and agrees with the changes made. 

Best regards,

David A. Fidock, Ph.D.

Guest Editor

PLOS Pathogens

Kami Kim

Section Editor

PLOS Pathogens

Kasturi Haldar

Editor-in-Chief

PLOS Pathogens

orcid.org/0000-0001-5065-158X

Michael Malim

Editor-in-Chief

PLOS Pathogens

orcid.org/0000-0002-7699-2064

The authors have provided an excellent rebuttal and revised manuscript, including additional experimentation as requested by Reviewer 2. That reviewer has gone through the revision and is satisfied by the response. This was also reviewed by a member of the Editorial team who is also satisfied by the changes. We are happy to recommend acceptance.
---

## [Editor Report · Acceptance letter]

5 Jun 2023

Dear Prof. Lanzer,

We are delighted to inform you that your manuscript, "PfCRT mutations conferring piperaquine resistance in falciparum malaria shape the kinetics of quinoline drug binding and transport," has been formally accepted for publication in PLOS Pathogens.

Best regards,

Kasturi Haldar

Editor-in-Chief

PLOS Pathogens

orcid.org/0000-0001-5065-158X

Michael Malim

Editor-in-Chief

PLOS Pathogens

orcid.org/0000-0002-7699-2064